# CAN GRAPH QUANTIZATION TOKENIZER CAPTURE TRANSFERABLE PARTTERNS?

## ABSTRACT

Graph tokenization aims to convert graph-structured data into discrete representations that can be used in foundation models. Recent methods propose to use vector quantization to map nodes or subgraphs into discrete token IDs. However, it remains unclear whether these quantized tokenizers truly capture high-level, transferable graph patterns across diverse domains. In this work, we conduct a comprehensive empirical study to analyze the representational consistency of quantized graph tokens across different datasets. We introduce the Graph Token Information Discrepency Score (GTID) to quantify the alignment of structural and feature information between source and target graphs for each token. Our results reveal that current graph quantized tokenizers often assign the same token to structurally inconsistent patterns across graphs, resulting in high GTID and degraded transfer performance. We further demonstrate that GTID is positively correlated with the generalization gap in downstream tasks. Finally, we propose a simple yet effective structural hard encoding (SHE) strategy to enhance the structural awareness of the tokenizer. SHE leads to lower GTID and improved transferability, highlighting the importance of explicitly encoding transferable graph structure in token design.

## 1 INTRODUCTION

In recent years, graph deep learning has emerged as a powerful toolkit for modeling data with inherent relational structures (Ma & Tang, 2021; Xia et al., 2021a; Wu et al., 2020). Unlike traditional data formats such as sequences (e.g., text) or grids (e.g., images), many real-world datasets, ranging from citation networks to molecules (Xia et al., 2023; Jumper et al., 2021) , can be naturally represented as graphs (Xia et al., 2021b). To effectively process graph-structured data, a variety of graph neural networks (GNNs) have been proposed, including Graph Convolutional Networks (Yao et al., 2019), Graph Attention Networks (Veličković et al., 2018), and Graph Transformers (Yun et al., 2019; Rampášek et al., 2022; Chen et al., 2022). These graph learning methods can model non-Euclidean data well and enable learning representations for nodes, edges, and entire graphs. However, despite the impressive success of GNNs in many tasks, they usually can be trained and applied to a single dataset. The efforts of the generalizing deep graph learning models to multiple datasets have only made limited progress due to the diversity and complexity of the graph data (Mao et al., 2024).

On the other hand, the success of foundation models (Brown et al., 2020; Achiam et al., 2023; Team et al., 2023) in natural language processing (NLP) and computer vision (CV) has motivated researchers to explore analogous approaches for graphs (Mao et al., 2024). One of the important attempts is graph tokenization (Yang et al., 2023; Chen et al., 2024a), a method inspired by text and image tokenization, where raw graph inputs are transformed into a sequence or set of "tokens" that can be processed by powerful sequence models like transformers. Just as words or subwords serve as basic units in language modeling, graph tokens aim to represent meaningful atomic or composite units of graph data.

However, unlike the tokens can be naturally defined in language, there are no obvious basic unit in the graph data. Hence, following the successful examples in CV (van den Oord et al., 2017; Lee et al., 2022b; Tian et al., 2024), researchers recently proposed the quantization graph tokenizer (Wang et al., 2024b; Luo et al., 2024) to learn the token representations. Specifically, the graph quantization tokenization will learn to convert a graph or subgraph into a set of vectorized representations (tokens) that encapsulate both the structural and feature information present in the original graph. Once the

tokenizer is trained, they can be applied to more datasets and generate graph tokens. Currently, the quantized graph tokenizer has achieved certain success in both supervised and unsupervised learning senarios and on different downstream tasks such as node classification, link prediction, or graph classification (Luo et al., 2024; Liu et al., 2023b; Wang et al., 2024a;b).

However, a fundamental question arises: *Do current graph tokenization methods actually capture the high-level, transferrable patterns inherent in graph data?* In other words, do the quantized tokens encode the vital graph structural information, instead of assigning the tokens heavily based on the *raw node features*?

This question is related to the fundamental capabilities of graph quantized tokenizers. Many downstream tasks in graph learning rely heavily on recognizing high-level structural patterns, such as degree distribution, homophily, and centrality. For example, in drug discovery, subtle topological variations in molecular graphs—captured by molecular topology and centrality descriptors—can directly influence biological activity and binding affinity (Zhang et al., 2025; Csermely et al., 2012). In social networks, tasks like community detection or influence modeling also depend critically on network connectivity and central nodes (Barabási & Oltvai, 2004; McPherson et al., 2001). When quantization tokenization fails to preserve these essential graph properties, the resulting graph tokens may omit meaningful structural patterns, impairing downstream task performance.

In this study, we present the first comprehensive empirical investigation into the knowledge encoded by graph quantized tokenizers. Specifically, we measure the discrepancy in both structural and feature information of nodes that are mapped to the same token across different datasets. Our analysis reveals that identical tokens often correspond to markedly different structural distributions in different graphs, indicating that *current graph quantized tokenizers fail to capture high-level, transferable patterns*. This deficiency undermines both the stability and cross-domain generalization ability of such tokenizers. The contributions of this work are as follows:

- We have analyzed both the structural and feature information encoded by the graph tokenizer. We find that there are significant information distribution discrepancies for the same token across different graphs.
- We show that the information discrepency of the tokens will hinder the model's transferrability, resulting in sub-optimal performance on the downstream tasks.
- Based on the findings above, we propose a trick to explicitly help the graph quantization tokenizer to encode the structural information. We show that the trick could mitigate the information discrepancy of tokens on different graphs, further affirming the value of our observations.

## 2 RELATED WORKS

Graph tokenization sits at the intersection of representation learning, graph neural networks (GNNs), and transformer-based models, drawing inspiration from tokenization practices in natural language processing and computer vision. Several strands of related research contribute to the development of tokenization methods for graph data.

**Quantization and Discrete Representation Learning.** Quantized latent representation learning has emerged as a powerful strategy to bridge the gap between continuous data and discrete symbolic reasoning. Among the most influential approaches, Vector Quantized Variational Autoencoder (VQ-VAE) (van den Oord et al., 2017; Esser et al., 2021) introduced a discrete bottleneck into the autoencoding framework, enabling learning of a codebook of latent embeddings that can compactly represent high-dimensional inputs. VQ-VAE has seen broad success in areas such as *image generation*, *speech modeling*, and *language modeling*, where discrete tokens enable autoregressive decoding and large-scale pretraining. Its extension, Residual Quantized VAE (RQ-VAE) (Lee et al., 2022a) addresses the limited capacity of shallow codebooks by employing *multi-level quantization*, decomposing inputs into multiple additive residuals. This yields richer token representations and better compression, making it particularly suitable for complex modalities.

**Quantized Representations in Graph Learning.** Despite the success of VQ-VAE in vision and language domains, its adaptation to *graph-structured data* remains relatively underexplored. Unlike pixels or words, graphs are *non-Euclidean* and *permutation-invariant*, posing significant challenges

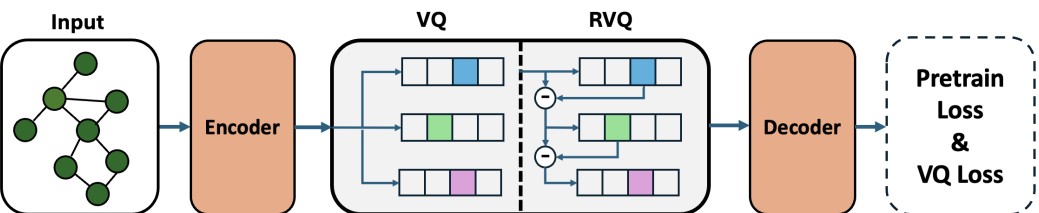

Figure 1: The pipeline of graph quantized tokenizer.

for tokenization. Several recent efforts have sought to bridge this gap. GraphMAE (Hou et al., 2022) and GPT-GNN (Hu et al., 2020) introduced self-supervised frameworks for node- and graph-level representation learning, but they rely on continuous encodings. A more direct attempt at tokenization can be found in GVT (You et al., 2023), which integrates VQ-VAE to learn discrete node prototypes and supports masked autoencoding on graphs. However, such methods typically apply quantization at the node level, ignoring higher-order structures or global subgraph semantics.

**Graph Pretraining with Structural Discreteness.** Recent works such as OneForAll (Liu et al., 2023a) and GFT (Graph Foundation Model with Transferable Tree Vocabulary) (Wang et al., 2024b) argue for discrete graph vocabulary learning to enable large-scale generalization across domains. OneForAll explores cross-domain pretraining with task-level tokenization, while GFT builds hierarchical tree vocabularies based on rooted subtrees, which are then quantized for structural reuse. Other notable approaches include AnyGraph (Xia & Huang, 2024), which aims to unify different graph modalities with plug-and-play architecture, and GraphPrompt (Jin et al., 2022), which leverages discrete prompts to guide downstream adaptation.

## 3 METHODOLOGY

In this section, we will introduce the graph quantized tokenizer to be investigated. We will first introduce the key components, namely Vector Quantization (VQ) and Residual Vector Quantization (RVQ). Next, we will introduce the whole pipeline as shown in Fig. 1.

### 3.1 VECTOR QUANTIZATION METHODS

**Vector Quantization (VQ)** (Gray, 1984; Gong et al., 2014; Esser et al., 2021) aims to represent a large set of vectors, $\boldsymbol{Z} = \{\boldsymbol{z}_i\}_{i=1}^N$, with a small set of prototype (code) vectors of a codebook $\boldsymbol{C} = \{\boldsymbol{e}_k\}_{k=1}^K$, where $N \gg K$. The codebook is often created using algorithms such as $k$-means clustering via optimizing the following objective:

$$\min_{\boldsymbol{C}} \sum_{i=1}^N \min_{k=1}^K ||\boldsymbol{z}_i - \boldsymbol{e}_k||_2^2. \tag{1}$$

Once the codebook is learned, each vector $\boldsymbol{z}_i$ can be approximated by its closet prototype vector $\boldsymbol{e}_t$, where $t = \arg\min_k ||\boldsymbol{z}_i - \boldsymbol{e}_k||_2^2$ is the index of the prototype vector.

**Residual Vector Quantization (RVQ)** (Juang & Gray, 1982; Martinez et al., 2014; Lee et al., 2022a) is an extension of the basic VQ. After performing an initial VQ, the *residual vector* is calculated:

$$\boldsymbol{r}_i = \boldsymbol{z}_i - \boldsymbol{e}_t, \tag{2}$$

which represents the quantization error from the initial quantization. Then, the residual vectors $\boldsymbol{r}_i$ are quantized using a second codebook. This process can be repeated multiple times, with each stage quantizing the residual error from the previous stage.

## 3.2 GRAPH QUANTIZED TOKENIZER

The graph quantized tokenizer intends to assign a token ID to the given node based on its own feature and neighboring nodes. To generate structure-aware node IDs, we employ an $L$-layer MPNN to capture multi-order neighborhood structures. At each layer, we use vector quantization to encode the node embeddings produced by the MPNN into $M$ codewords (integer indices). For each node $v$, we define the node ID of $v$ as a tuple composed of $L \times M$ codewords, structured as follows:

$$\text{Node\_ID}(v) = (c_{11}, \cdots, c_{1M}, c_{21}, \cdots, c_{2M}, \cdots) \tag{3}$$

where $c_{lm}$ represents the $m$-th codeword at the $l$-th layer. Both $M$ and $L$ are integers.

As illustrated in Fig. 1, at each layer $l$ ($1 \leq l \leq L$) of the MPNN, we employ VQ/RVQ to quantize the node embeddings and produce $M$ digits of codewords for each node $v$. Each codeword $c_{lm}$ ($1 \leq m \leq M$) is generated by a distinct codebook $\boldsymbol{C}_{lm} = \{\boldsymbol{e}_k^{lm}\}_{k=1}^K$, where $K$ is the size of the codebook. Hence, there are a total of $L \times M$ codebooks, indexed by $lm$. Let $\boldsymbol{r}_{lm}$ denote the vector to be quantized. Note that $\boldsymbol{r}_{l1}$ is the node embedding $\boldsymbol{h}_v^l$ produced by the MPNN. When $m > 1$, $\boldsymbol{r}_{lm}$ represents the residual vector. Then, $\boldsymbol{r}_{lm}$ is approximated by its nearest code vector from the corresponding codebook $\boldsymbol{C}_{lm}$:

$$c_{lm} = \arg\min_k \|\boldsymbol{r}_{lm} - \boldsymbol{e}_k^{lm}\|, \tag{4}$$

producing the codeword $c_{lm}$, which is the index of the nearest code vector.

We follow the existing framework for learning node token IDs (codewords $c_{lm}$) by jointly training the MPNN and the codebooks with the following loss function:

$$\mathcal{L}_{\text{total}} = \mathcal{L}_\mathcal{G} + \mathcal{L}_{\text{VQ}}, \tag{5}$$

where $\mathcal{L}_\mathcal{G}$ is a (self-)supervised graph learning objective, and $\mathcal{L}_{\text{VQ}}$ is a vector quantization loss. $\mathcal{L}_\mathcal{G}$ aims to train the MPNN to produce effective node embeddings, while $\mathcal{L}_{\text{VQ}}$ ensures the codebook vectors align well with the node embeddings. For a single node $v$, $\mathcal{L}_{\text{VQ}}$ is defined as

$$\mathcal{L}_{\text{VQ}} = \sum_{l=1}^L \sum_{m=1}^M \|\text{sg}(\boldsymbol{r}_{lm}) - \boldsymbol{e}_{c_{lm}}^{lm}\| + \beta\|\boldsymbol{r}_{lm} - \text{sg}(\boldsymbol{e}_{c_{lm}}^{lm})\|, \tag{6}$$

where sg denotes the stop gradient operation, and $\beta$ is a weight parameter. The first term in Eq. (6) is the *codebook loss*, which only affects the codebook and brings the selected code vector close to the node embedding. The second term is the *commitment loss*, which only influences the node embedding and ensures the proximity of the node embedding to the selected code vector. In practice, we can use exponential moving averages (Razavi et al., 2019) as a substitute for the *codebook loss*.

The graph learning objective $\mathcal{L}_\mathcal{G}$ can be a self-supervised learning task, such as graph reconstruction (i.e., reconstructing the node features or graph structures) or contrastive learning (Liu et al., 2021). In this paper, we follow most of the existing works that utilize GraphMAE (Hou et al., 2022). GraphMAE involves sampling a subset of nodes $\tilde{\mathcal{V}} \subset \mathcal{V}$, masking the node features as $\tilde{\boldsymbol{X}}$, encoding the masked node features using an MPNN, and subsequently reconstructing the masked features with a decoder. The reconstruction loss is based on the scaled cosine error, expressed as:

$$\mathcal{L}_{\text{MAE}} = \frac{1}{|\tilde{\mathcal{V}}|} \sum_{v \in \tilde{\mathcal{V}}} \left(1 - \frac{\boldsymbol{x}_v^T \boldsymbol{z}_v}{\|\boldsymbol{x}_v\| \cdot \|\boldsymbol{z}_v\|} \cdot \gamma\right),$$

where $\tilde{\mathcal{V}}$ is the set of masked nodes, $\boldsymbol{z}_v = f_D(\tilde{\boldsymbol{h}}_v^L)$ is the reconstructed node features by a decoder $f_D$, $\tilde{\boldsymbol{h}}_v^L = \text{MPNN}(v, \boldsymbol{A}, \tilde{\boldsymbol{X}})$, and $\gamma \geq 1$ is a scaling factor. Let $\tilde{\boldsymbol{r}}_{l1} := \tilde{\boldsymbol{h}}_v^l$ denote the node embedding generated by the $l$-th layer of the MPNN with the masked features. The overall training loss is

$$\mathcal{L}_{\text{total}} = \mathcal{L}_{\text{MAE}} + \sum_{v \in \tilde{\mathcal{V}}} \sum_{l=1}^L \sum_{m=1}^M \|\text{sg}(\tilde{\boldsymbol{r}}_{lm}) - \boldsymbol{e}_{c_{lm}}\| + \beta\|\tilde{\boldsymbol{r}}_{lm} - \text{sg}(\boldsymbol{e}_{c_{lm}})\|. \tag{7}$$

# 4 PRELIMILARY

## 4.1 EXPERIMENT SETUPS

Here we first introduce our experiment setups, i.e., how we train and evaluate the graph quantized tokenizer. In order to obtain the comprehensive results, we train and evaluate both VQ and RVQ methods. For all the models, we set the number of MPNN layers to be 2, and the number of codewords to be 3. We train the tokenizer on the datasets from two domains: citation graphs and e-commerce networks. The citation graphs include: cora, citesser, dblp, arxiv and pubMed. The e-commerce graphs include: bookhis, bookchild, elecomp, elephoto and sportsfit. The detailed information of the datasets can be found in Appendix A. For each domain, we pretrain the tokenzier on 1 to 4 datasets and then use infer on the remaining datasets in the domain. On both training and test datasets, we will record the subgraphs that assigned to each token ID. For instance, for a node token ID $c_{mn}$, we will record the subgraphs in training set assigned to it as a $S_{mn,train}$, and we will record the the subgraphs in test set assigned to it as a $S_{mn,test}$ Then we would calculate the information discrepancy between $S_{mn,train}$ and $S_{mn,test}$ for each token.

## 4.2 EVALUATION METRIC

Here we will introduce the metric we design to measure the information discrepancy of tokens. Specifically, we design a metric named Graph Token Information Discrepency Score (GTID) to calculate the discrepancy between $S_{mn,train}$ and $S_{mn,test}$. Suppose the representations of $S_{mn,train}$ and $S_{mn,tes}$ are $f_{mn,train}$ and $f_{mn,test}$. Following the previous works (Yan et al., 2017; Wang et al., 2021), we use Maximum Mean Discrepancy (MMD) to calculate the discrepancy between $f_{mn,train}$ and $f_{mn,test}$. Specifically, we tend to compare the MMD computed on node features and structures. Therefore, we adapt Normalized Maximum Mean Discrepancy (NMMD) in this work. First we normalize the vectors in $f_{mn,train}$ and $f_{mn,test}$ and denote $\hat{f}_{mn,train} = \{\mathbf{p}_i\}_{i=1}^v$ and $\hat{f}_{mn,test} = \{\mathbf{q}_i\}_{i=1}^w$: Then we first calculate the MMD of the two vector sets:

$$\widehat{\text{MMD}}^2 = \frac{1}{v^2}\sum_{i=1}^v\sum_{i'=1}^v k(\mathbf{p}_i,\mathbf{p}_{i'}) + \frac{1}{w^2}\sum_{j=1}^w\sum_{j'=1}^w k(\mathbf{q}_j,\mathbf{q}_{j'}) - \frac{2}{vw}\sum_{i=1}^v\sum_{j=1}^w k(\mathbf{p}_i,\mathbf{q}_j). \tag{8}$$

where $k(\cdot,\cdot)$ is an RKHS kernel. And next we calculate the variance-normalized MMD:

$$\widehat{\text{NMMD}}^2 = \frac{\widehat{\text{MMD}}^2}{\widehat{V}}, \qquad \widehat{V} = \frac{1}{v}\sum_{i=1}^v k(\mathbf{p}_i,\mathbf{q}_i) + \frac{1}{w}\sum_{j=1}^w k(\mathbf{q}_j,\mathbf{q}_j). \tag{9}$$

Finally, the GTID between the train and test domains is calculated with average of the normalized maximum mean discrepancy on all the codewords:

$$\text{GTID} = \frac{\sum_m\sum_n \text{NMMD}(f_{mn,train}, f_{mn,test})}{mn} \tag{10}$$

The more information of calculation of Maximum Mean Discrepancy and Normalized Maximum Mean Discrepancy can be found in the Appendix. In general, the larger GTID is, the larger information discrepancy is.

Since the subgraphs contain both structural and feature information, we will calculate the GTID for node features and structures respectively. For node features, we adapt the set of center node features as $f_{mn,train}$ and $f_{mn,test}$. For the structures, we calculate the structure property vectors [degree, clustering coefficient, closeness centrality, density, assortativity, transitivity, homophily] as the representations. We give the details of calculating the structural properties in Appendix C. Next, we will analyze the GTID and their relations to the model generalization. We observed similar phenomena for both RVQ and VQ tokenizers. Hence, we mainly discuss the results based on RVQ tokenizer and leave the results of VQ tokenizer to Appendix E.

## 4.3 THEORETICAL ANALYSIS

Before diving into the empirical observations, we first derive theorectical analysis to prove the relationship between the token information discrepancy and the model transferability. We tend to prove that low information discrepancy in tokens can lead to higher transferability and do this for both node features and ego-graph structures. We will first give the theorems and provide the full proof in Appendix D.

**Theorem 1** (Code-Conditional Transfer Bound: Feature View). *Let $\mathcal{D}_s, \mathcal{D}_t$ be source/target node datasets (from graphs $G_s, G_t$). Each node $v$ has an $L$-hop ego-subgraph $g(v)$ with feature tensor. A fixed encoder $\phi : \mathcal{G} \to \mathbb{R}^m$ maps $g$ to $z = \phi(g)$. A codebook $Q$ with codes $\{c_1, \ldots, c_K\}$ assigns $J(g) = Q(z) \in [K]$ by nearest center. A predictor $h$ consumes a feature summary $u(g) \in \mathbb{R}^{d'}$ (e.g., pooled/root features), and the loss $\ell : \mathcal{Y} \times \mathcal{Y} \to [0, 1]$ is bounded.*

***Notation.*** *Let $\pi_\alpha(k) = \Pr_{(g,y) \sim \mathcal{D}_\alpha}[J(g) = k]$ for $\alpha \in \{s, t\}$, and let $\mathbb{P}_\alpha^k$ be the conditional law of $(g, x, y)$ given $K(g) = k$. Define risks $\varepsilon_\alpha(h \circ Q \circ \phi) = \mathbb{E}_{\mathcal{D}_\alpha}[\ell(h(Q(\phi(g))), y)]$. Let the code-marginal drift be $\Delta_{\mathrm{code}} := \frac{1}{2} \sum_{k=1}^{K} |\pi_t(k) - \pi_s(k)|$. Define the quantization distortion $\Delta_{\mathrm{q}} := \sup_{(g,y)} |\ell(h(Q(\phi(g))), y) - \ell(h(\phi(g)), y)|$, and let $p_{\mathrm{mis}} := \Pr[K(g) \text{ is a misassignment}]$ (e.g., stale codebook/ANN search).*

***Assumptions.*** *(i) There exists a (possibly identity) preprocessing $S : \mathbb{R}^{d'} \to \mathbb{R}^{d'}$ such that $u \mapsto \ell(h(u), y)$ is $L_u$-Lipschitz uniformly in $y$. (ii) For each code $k$, define the within-code feature discrepancy*

$$\Delta_k^{\mathrm{feat}} := W_1\big(\mathcal{L}_t(S(u) \mid K = k), \mathcal{L}_s(S(u) \mid K = k)\big),$$

*the 1-Wasserstein distance between the conditional feature summaries.*

***Claim.*** *For any $\delta \in (0, 1)$, with probability at least $1 - \delta$ over the draws of the (finite) datasets and the code-conditional estimates,*

$$\varepsilon_t - \varepsilon_s \leq \sum_{k=1}^{K} \pi_t(k) L_u \Delta_k^{\mathrm{feat}} + \Delta_{\mathrm{code}} + \Delta_{\mathrm{q}} + p_{\mathrm{mis}} + c_1 \sqrt{\frac{\log(2K/\delta)}{\min_k n_t(k)}} + c_2(\mathfrak{R}_{n_s} + \mathfrak{R}_{n_t})$$

*Here $n_t(k)$ is the number of target samples with $K = k$, $\mathfrak{R}_{n_\alpha}$ denotes the Rademacher complexity of the induced loss class on domain $\alpha$, and constants $c_1, c_2$ depend only on sub-Gaussian/boundedness parameters of $S(u)$ and on standard symmetrization constants.*

**Theorem 2** (Code-Conditional Transfer Bound: Structure View). *Same as Theorem 1, except the predictor $h$ depends on a structural representation $\psi(g)$ that lies in an RKHS $(\mathcal{H}, \langle \cdot, \cdot \rangle_\mathcal{H})$ with kernel $k(\cdot, \cdot)$ and $\|\psi(g)\|_\mathcal{H} \leq B$. Risks, $\pi_\alpha(k)$, $\mathbb{P}_\alpha^k$, $\Delta_{\mathrm{code}}$, $\Delta_{\mathrm{q}}$, and $p_{\mathrm{mis}}$ are as defined there.*

***Assumptions.*** *(i) For each code $k$, the conditional loss as a function of $\psi(g)$ belongs to a bounded RKHS ball: there exists $f_k \in \mathcal{H}$ with $\|f_k\|_\mathcal{H} \leq C$ such that $\mathbb{E}[\ell(h(\psi(g)), y) \mid g, K = k] = \langle f_k, \psi(g) \rangle_\mathcal{H}$. (ii) For each code $k$, define the within-code structural discrepancy*

$$\Delta_k^{\mathrm{struct}} := \mathrm{MMD}_\mathcal{H}\big(\mathcal{L}_t(\psi \mid K = k), \mathcal{L}_s(\psi \mid K = k)\big).$$

***Claim.*** *For any $\delta \in (0, 1)$, with probability at least $1 - \delta$,*

$$\varepsilon_t - \varepsilon_s \leq \sum_{k=1}^{K} \pi_t(k) C \Delta_k^{\mathrm{struct}} + \Delta_{\mathrm{code}} + \Delta_{\mathrm{q}} + p_{\mathrm{mis}} + \tilde{c}_1 \sqrt{\frac{\log(2K/\delta)}{\min_k n_t(k)}} + \tilde{c}_2(\mathfrak{R}_{n_s} + \mathfrak{R}_{n_t}),$$

*where $\tilde{c}_1, \tilde{c}_2$ depend only on the kernel bound $k(x, x) \leq B^2$ and standard generalization constants.*

## 5 RESULTS AND OBSERVATIONS

### 5.1 THE GTID DISTRIBUTIONS

Following the evaluation process above, we can pretrain the tokenizers and evaluate them. Specifically, we would pretrain the tokenizer with different combinations of datasets: from single dataset to four

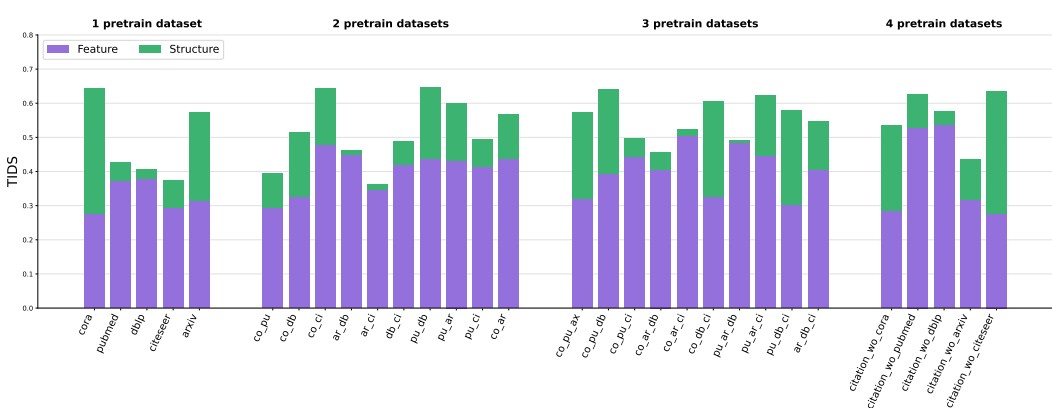

Figure 2: The distributions of GTID of RVQ models on citation datasets. We use abbreviated names for datasets in x-axis. co: cora, pu: pubmed, db: dblp, ci: citeseer, ar: arxiv.

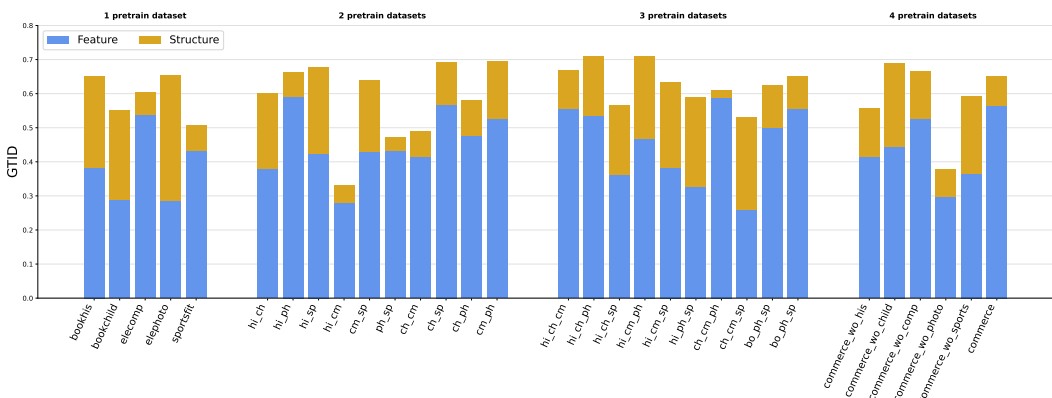

Figure 3: The distributions of GTID of RVQ models on e-commerce datasets. We use abbreviated names for datasets in x-axis. hi: bookhis, ch: bookchild, cm: elecomp, ph: elephoto, sp: sportsfit.

datasets together. Then we will evaluate them on the remaining datasets in the same domain and calculate the corresponding GTID. The results are shown in Figure 2 and Figure 3 for the Citation domain and E-commerce domain, respectively.

Across both domains, we observe a consistent and obvious gap between structure-based and feature-based GTID. While feature discrepancy tends to decrease gradually as the number of pretraining datasets increases, the structural GTID remains relatively high and fluctuates across settings. This suggests that even with multi-dataset pretraining, the tokenizer struggles to align structural information consistently. For instance, in the Citation domain (Figure 2), structural GTID plateaus after the second pretraining dataset, indicating limited marginal gains in structural transferability. A similar trend is seen in the E-Commerce domain where feature-based discrepancy steadily decreases but structural discrepancy remains elevated, particularly in dataset groups that are structurally diverse.

Furthermore, while tokenizers benefit from more diverse feature distributions during pretraining, their ability to generalize structural semantics is far more constrained. This asymmetry highlights a key limitation of current quantization-based tokenizers: their reliance on local node features or first-order neighborhoods makes it difficult to internalize structural motifs that generalize across domains with heterogeneous graph topology. Hence, we would have the following observation:

**Observation 1: The graph quantized tokenizes have difficulty capturing the transferrable patterns across graphs, especially the structural patterns.**

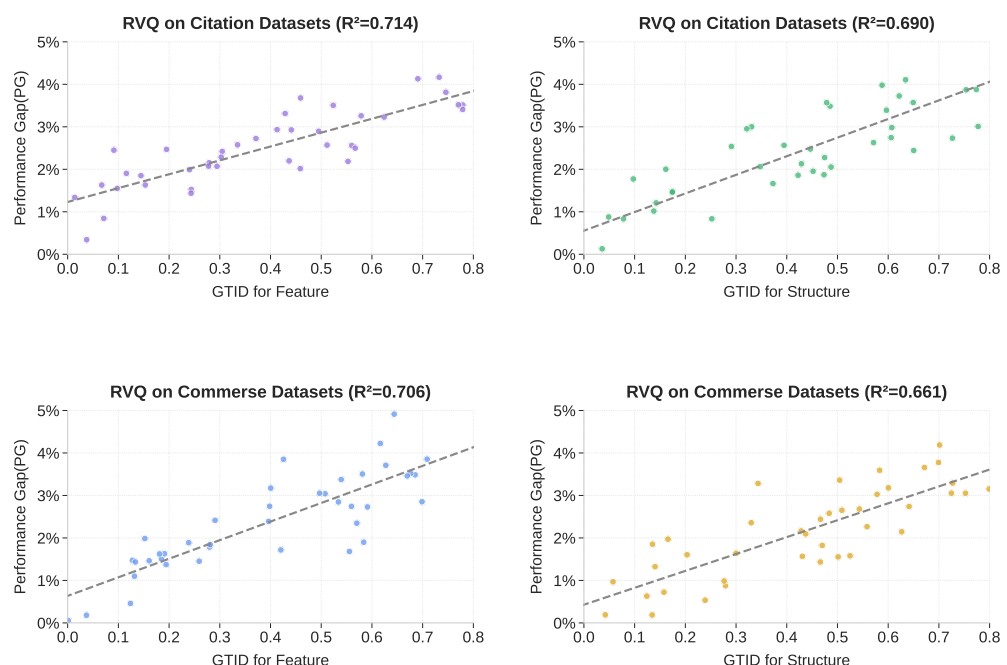

Figure 4: The correlation between GTID and the model performance gap.

## 5.2 THE CORRELATIONS BETWEEN GTID AND MODEL'S TRANSFERABILITY

Furthermore, we leverage the results above to analyze the relationship between model generalization and performance gaps. Specifically, we define the *performance gap* (PG) as a metric to quantify generalization ability, measured by the accuracy difference between inter-dataset and intra-dataset pretraining.

For example, consider two datasets, $A$ and $B$. Let $P_1$ denote the node classification accuracy of a model pretrained on $A$ and fine-tuned on $B_{\text{train}}$, and $P_2$ denote the accuracy of a model both pretrained and fine-tuned on $B_{\text{train}}$ ($B_{\text{train}}$ and $B_{\text{train}}$ are the training part and test part of dataset B, respectively). The performance gap is then computed as:

$$\text{PG} = \frac{P_2 - P_1}{P_2}.$$

This normalized gap reflects how well the pretrained knowledge transfers across datasets. The results are shown in Figure 4. The reported *coefficient of determination* ($R^2$) quantifies the extent to which GTID explains the transfer performance degradation.

The results are shown in Figure 4. Across all settings, we observe a strong positive correlation between GTID and performance gap. In the Citation domain, feature-based GTID achieves an $R^2$ of 0.714, while structure-based GTID yields 0.707. A similar trend is observed in the E-Commerce domain, where the feature and structure correlations yield $R^2$ values of 0.709 and 0.692, respectively. These results suggest that both forms of token discrepancy significantly affect downstream transferability, with feature discrepancy often exhibiting slightly higher explanatory power, potentially due to its stronger alignment with task-relevant attributes.

These findings indicate that token consistency across domains is critical for effective transfer learning. When the same token index encodes semantically or structurally divergent patterns across graphs, the transfer model struggles to leverage pre-learned representations. This mismatch leads to notable performance degradation during cross-domain adaptation.

**Observation 2: The GTID is positively correlated the performance gap, indicating that the information discrepancy of the tokens will hinder model's transferability.**

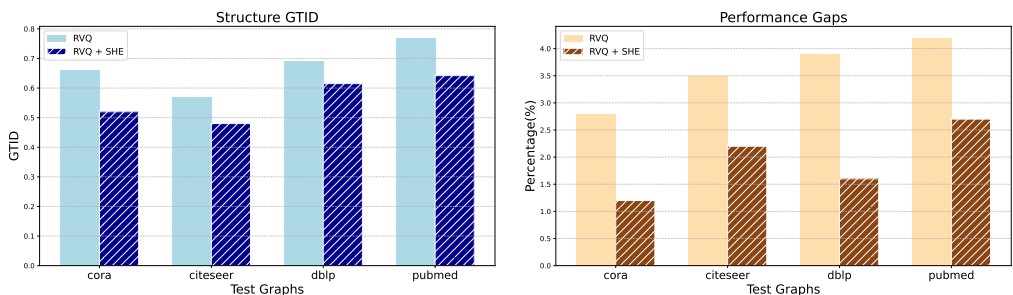

Figure 5: Comparison with the original RVQ tokenizer after utilizing the structural hard encoding.

## 5.3 Structural Hard Encoding

To evaluate whether adding structural information to the tokenizer can improve transferability, we incorporate a simple yet effective inductive bias: **Structural Hard Encoding (SHE)**. SHE explicitly encodes high-level structural cues (e.g., node degree bins, positional encodings) into the input of the quantized tokenizer, aiming to reduce the mismatch in structural semantics across graphs. For instance, nodes with degree 1 or 2 can only be assigned to ID 0 to 31, nodes with degree 3 can only assigned to ID 32 to 63, etc. In this way, we force the token ID distinguish with each other as their corresponding subgraphs will have structural properties' differece.

As shown in Figures 5, SHE leads to a notable improvement in both structural alignment and downstream task performance. In Figure 5 Left, we observe that for all test graphs (Cora, Citeseer, DBLP, Pubmed), the *structure-based GTID* is consistently lower when using RVQ with SHE compared to vanilla RVQ. This reduction is especially pronounced on datasets with higher structural variability (e.g., DBLP and Pubmed), indicating that SHE effectively mitigates token inconsistency arising from structural heterogeneity.

The benefits of this structural regularization also translate into improved model generalization. Figure 5 Right shows that the *performance gap* between source-pretrained and target-finetuned models is also reduced across the same set of graphs when SHE is applied. This reinforces the claim that lower GTID correlates with improved transferability, and affirms that enhancing structural awareness during tokenization is a viable pathway to better cross-graph generalization. Hence, we would have the following observation:

**Observation 3: With structural hard encoding (SHE), the RVQ tokenizer can reduce the structural GTID and performance gaps, which further affirm our previous observations and the importance of capturing transferrable for tokens.**

## 6 Conclusion

In this paper, we investigate whether graph quantized tokenizers can capture transferable patterns across graph datasets. Through a detailed empirical analysis, we show that tokenized representations suffer from significant information discrepancies, particularly in structural properties, across different domains. We introduce the Token Information Discrepancy Score (TIDS) to quantify this phenomenon and demonstrate its strong correlation with performance degradation in transfer learning settings. These findings indicate that current quantized tokenization schemes are limited in their ability to produce consistent, reusable representations for graph data. To address this, we propose Structural Hard Encoding (SHE), a simple inductive bias that explicitly incorporates structural signals into the token assignment process. Our experiments show that SHE significantly reduces structural TIDS and improves cross-domain performance, validating our core hypothesis. This work provides actionable insights into the limitations of current graph tokenizers and opens up future research directions on structure-aware, transferable graph token learning.

ETHICS STATEMENT

We acknowledge that we have read and commit to adhering to the ICLR Code of Ethics. Our study relies solely on publicly available benchmark datasets Chen et al. (2024b). While our proposed method presents no direct ethical concerns, the improved performance could be leveraged for both ethical and unethical applications involving generative recommendation systems. We emphasize the importance of applying machine learning algorithms responsibly to achieve socially beneficial results.

REPRODUCIBILITY STATEMENT

Our experiments are based on the public datasets and code (Wang et al., 2024b; Chen et al., 2024b). To help reproducibility of the results, we provide experiment settings in the main text.

USAGE OF LARGE LANGUAGE MODELS

In this manuscript, we solely utilize LLMs to polish the writing and check grammatical errors. We have reviewed the generated contents provided by large language models and will be responsible for the correctness of the polished content.

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

## A DATASET DETAILS

Table 1 presents the detailed statistics of datasets we used in our experiments, including the dataset's domain and sizes.

Table 1: Dataset statistics.

| Dataset | Domain | # Nodes | # Edges |
|---|---|---|---|
| Cora | Citation | 2708 | 10556 |
| Citeseer | Citation | 3186 | 8450 |
| Pubmed | Citation | 19717 | 88648 |
| DBLP | Citation | 14376 | 431326 |
| Arxiv | Citation | 169343 | 2315598 |
| Bookhis | E-commerce | 41551 | 503180 |
| Bookchild | E-commerce | 76875 | 2325044 |
| Elecomp | E-commerce | 87229 | 1256548 |
| Elephoto | E-commerce | 48362 | 873782 |
| Sportsfit | E-commerce | 173055 | 3020134 |

## B MAXIMUM MEAN DISCREPANCY

Maximum Mean Discrepancy (MMD) is a statistical distance metric used to measure the discrepancy between two probability distributions $P$ and $Q$ over a domain $\mathcal{X}$. Formally, given a function class $\mathcal{F}$, the MMD is defined as

$$\text{MMD}[\mathcal{F}, P, Q] = \sup_{f \in \mathcal{F}} \left( \mathbb{E}_{x \sim P}[f(x)] - \mathbb{E}_{y \sim Q}[f(y)] \right).$$

When $\mathcal{F}$ is chosen to be the unit ball in a Reproducing Kernel Hilbert Space (RKHS) $\mathcal{H}$ with kernel function $k$, the squared MMD can be computed in closed form as

$$\text{MMD}^2(P, Q) = \mathbb{E}_{x, x' \sim P}[k(x, x')] + \mathbb{E}_{y, y' \sim Q}[k(y, y')]$$

For empirical distributions based on samples $\{x_i\}_{i=1}^{m}$ from $P$ and $\{y_j\}_{j=1}^{n}$ from $Q$, an unbiased estimator of the squared MMD is given by

$$\text{MMD}^2(P, Q) = \frac{1}{m(m-1)} \sum_{i \neq j} k(x_i, x_j) + \frac{1}{n(n-1)} \sum_{i \neq j} k(y_i, y_j)$$

This formulation makes MMD particularly useful for two-sample tests and as a loss function in machine learning tasks such as domain adaptation and generative modeling. The Normalized Maximum Mean Discrepancy is calculated as

$$\text{Normalized\_MMD}^2(P, Q) = \frac{\text{MMD}^2(P, Q)}{\text{MMD}^2(P, P) + \text{MMD}^2(Q, Q)}$$

## C THE DEFINATIONS OF THE STRUCTURAL PROPERTIES

**Degree (node & average)**

$$k_i = \sum_{j=1}^{n} A_{ij}, \qquad \bar{k} = \frac{1}{n} \sum_{i=1}^{n} k_i = \frac{2m}{n}.$$

**Local clustering coefficient & global averages**:

$$C_i = \begin{cases} \dfrac{2\, t_i}{k_i(k_i - 1)}, & k_i \geq 2, \\ 0, & k_i < 2, \end{cases} \quad \text{where} \quad t_i = \sum_{1 \leq p < q \leq n} A_{ip} A_{iq} A_{pq}.$$

$$C_{\text{avg}} = \frac{1}{n} \sum_{i=1}^{n} C_i.$$

**Closeness centrality**:

$$\text{clo}(i) = \sum_{\substack{j=1 \\ j \neq i}}^{n} d(i,j), \qquad \text{CC}(i) = \frac{n-1}{\text{clo}(i)}.$$

**Density**:

$$\delta(G) = \frac{2m}{n(n-1)}.$$

**Degree assortativity**: Let $\mu = \frac{1}{2m} \sum_{(u,v)\in E} (k_u + k_v)$.

$$r_{\text{deg}} = \frac{\frac{1}{m} \sum_{(u,v)\in E} k_u k_v - \mu^2}{\frac{1}{m} \sum_{(u,v)\in E} \frac{k_u^2 + k_v^2}{2} - \mu^2}.$$

**Transitivity**:

$$T = \frac{3\triangle}{\wedge} = \frac{\sum_{i=1}^{n} 2\,t_i}{\sum_{i=1}^{n} k_i(k_i - 1)},$$

where $\triangle$ is the number of triangles and $\wedge = \sum_i \binom{k_i}{2}$ is the number of connected triples.

**Homophily**: Given a discrete node attribute $x : V \to \{1, \ldots, C\}$, define

$$H_{\text{edge}} = \frac{1}{m} \sum_{(u,v)\in E} \mathbf{1}[x(u) = x(v)] \quad \text{(edge homophily rate)}.$$

Let $p_c = \frac{|\{i \in V : x(i) = c\}|}{n}$ and $H_0 = \sum_{c=1}^{C} p_c^2$. A normalized (chance-corrected) homophily index is

$$H_{\text{norm}} = \frac{H_{\text{edge}} - H_0}{1 - H_0}.$$

## D  PROOF FOR THE THEOREMS

Since the two theorems have similar structures, we will prove them parallely in this section. We will first introduce some definations and notations and will then move to the proof.

**Setting.** Let $\mathcal{D}_s, \mathcal{D}_t$ be source/target node datasets drawn from graphs $G_s, G_t$, respectively. Each node $v$ has an $L$-hop ego-subgraph $g(v)$ with feature tensor; let $\phi : \mathcal{G} \to \mathbb{R}^m$ be a (fixed) encoder, and $Q$ a codebook with codes $\{c_1, \ldots, c_K\}$. Write $Z = \phi(g)$ and $J(g) = Q(Z) \in [K]$. A predictor $h$ maps either (i) a *feature summary* $u(g) \in \mathbb{R}^{d'}$ or (ii) a *structural embedding* $\psi(g) \in \mathcal{H}$ to a prediction; the loss $\ell$ is bounded in $[0,1]$.

Let $\pi_\alpha(k) = \mathbb{P}_{(g,y)\sim\mathcal{D}_\alpha}[J(g) = k]$ for $\alpha \in \{s, t\}$, and $\mathbb{P}_\alpha^k$ be the law of $(g, x, y)$ conditional on $K(g) = k$. Define risks $\varepsilon_\alpha(h \circ Q \circ \phi) = \mathbb{E}_{\mathcal{D}_\alpha} \ell(h(Q(\phi(g))), y)$.

We also consider the *pre-quantization predictor* $\tilde{f} = h \circ \phi$ and the *post-quantization predictor* $f = h \circ Q \circ \phi$. Define the *quantization distortion*

$$\Delta_{\text{q}} := \sup_{(g,y)} |\ell(h(Q(\phi(g))), y) - \ell(h(\phi(g)), y)|.$$

Let $M(g)$ be the event that $g$ is assigned to a code whose center lies outside a radius-$\tau$ cell around $\phi(g)$ (misassignment due to finite codebook update); set $p_{\mathrm{mis}} = \mathbb{P}[M(g)]$.

**Reproducing Kernel Hilbert Space (RKHS) setting.** Let $(\mathcal{H}, k)$ be a reproducing kernel Hilbert space associated with a positive-definite kernel $k : \mathcal{G} \times \mathcal{G} \to \mathbb{R}$. Typical choices include graph kernels such as

- the Weisfeiler–Lehman subtree kernel,
- the shortest-path or random-walk kernel,
- or a message-passing neural kernel $k(g, g') = \langle \psi(g), \psi(g') \rangle$ where $\psi(g)$ is the feature map of a base GNN layer.

We assume $k$ is bounded, $k(g, g) \leq B^2$, and that the feature map $\psi(g)$ satisfies $\|\psi(g)\|_{\mathcal{H}} \leq B$ for all subgraphs $g$. This ensures that $\mathrm{MMD}_{\mathcal{H}}$ is well-defined and bounded in $[0, 2B]$. Specifically, we use Let $A \in \mathbb{R}^{n \times d}$ and $B \in \mathbb{R}^{m \times d}$. Denote the $i$-th row of $A$ by $a_i \in \mathbb{R}^d$ and the $j$-th row of $B$ by $b_j \in \mathbb{R}^d$. The Gaussian (RBF) kernel matrix $K \in \mathbb{R}^{n \times m}$ with bandwidth $\sigma > 0$ is defined entrywise as

$$K_{ij} = \exp\left( -\frac{\|a_i - b_j\|_2^2}{2\sigma^2} \right), \qquad i = 1, \ldots, n,\ j = 1, \ldots, m.$$

**Code-wise discrepancy metrics.** For each code $k$:

*(Feature view)* Fix a 1-Lipschitz map $S : \mathbb{R}^{d'} \to \mathbb{R}^{d'}$ (possibly identity) and suppose the composed map $u \mapsto \ell(h(u), y)$ is $L_u$-Lipschitz uniformly in $y$. Let

$$\Delta_k^{\mathrm{feat}} := W_1\big( \mathcal{L}_t(S(u) \mid k),\, \mathcal{L}_s(S(u) \mid k) \big).$$

*(Structure view)* Let $\psi(g) \in \mathcal{H}$ be a bounded kernel embedding with $\|\psi(g)\|_{\mathcal{H}} \leq B$; assume the function $f_\psi : \mathcal{H} \to [0, 1]$ defined by $f_\psi(\psi(g)) = \mathbb{E}[\ell(h(\psi(g)), y) \mid g]$ lies in the RKHS ball $C\mathbb{B}_{\mathcal{H}}$. Define

$$\Delta_k^{\mathrm{struct}} := \mathrm{MMD}_{\mathcal{H}}\big( \mathcal{L}_t(\psi \mid k),\, \mathcal{L}_s(\psi \mid k) \big).$$

Additionally define the *code-marginal drift*

$$\Delta_{\mathrm{code}} := \mathrm{TV}(\pi_t, \pi_s) = \frac{1}{2} \sum_{k=1}^{K} |\pi_t(k) - \pi_s(k)|.$$

**Loss class and calibration.** Let $\mathcal{F} = \{ g \mapsto \ell(h(\cdot), y) \}$ be the induced loss class after $u$ or $\psi$. Assume a *margin-calibrated* property: there exists a non-decreasing $\Gamma : [0, 1] \to [0, 1]$ s.t. $|\mathbb{E}_{\mathbb{P}} f - \mathbb{E}_{\mathbb{Q}} f| \leq \Gamma\big(\mathrm{IPM}(\mathbb{P}, \mathbb{Q})\big)$ for $f \in \mathcal{F}$, where $\mathrm{IPM} = W_1$ in the feature case, and $\mathrm{IPM} = \mathrm{MMD}_{\mathcal{H}}$ in the structure case. For Lipschitz/$\mathcal{H}$-bounded classes we can take $\Gamma(r) = L_u r$ and $\Gamma(r) = Cr$, respectively.

**Finite-sample estimation.** Suppose we observe $n_\alpha$ i.i.d. nodes from $\mathcal{D}_\alpha$, with $n_\alpha(k)$ landing in code $k$. Let $\widehat{\Delta}_k^{\mathrm{feat}}$ (resp. $\widehat{\Delta}_k^{\mathrm{struct}}$) be empirical estimators. Assume $S(u)$ is sub-Gaussian with proxy $\sigma^2$ (per coordinate), and the kernel for $\psi$ is bounded by $B$. Let $\delta \in (0, 1)$.

**Theorem.** With probability at least $1 - \delta$, simultaneously for the feature and structure views,

$$\varepsilon_t(f) - \varepsilon_s(f) \leq \underbrace{\sum_{k=1}^{K} \pi_t(k)\, \Gamma\big(\Delta_k\big)}_{\text{code-conditional shift}} + \Delta_{\mathrm{code}} + \Delta_{\mathrm{q}} + p_{\mathrm{mis}}$$
$$+ \underbrace{c_1 \sqrt{\frac{\log(2K/\delta)}{\min_k n_t(k)}}}_{\text{conditional estimation}} + \underbrace{c_2\Big( \mathfrak{R}_{n_s}(\mathcal{F}) + \mathfrak{R}_{n_t}(\mathcal{F}) \Big)}_{\text{function class complexity}},$$

where $\Delta_k$ equals $\Delta_k^{\mathrm{feat}}$ in the feature view (with $\Gamma(r) = L_u r$) and equals $\Delta_k^{\mathrm{struct}}$ in the structure view (with $\Gamma(r) = Cr$). Constants $c_1, c_2$ depend only on universal sub-Gaussian/kernel bounds.

**Remarks.** (i) The first three additive terms quantify, respectively, *within-code* conditional mismatch, *code-marginal* mismatch, and *quantization* distortion; $p_{\mathrm{mis}}$ captures assignment noise (e.g., stale codebook). (ii) The last two terms are finite-sample effects: conditional-IPM estimation error and richness of the induced loss class. (iii) If $\phi$ is $L_\phi$-Lipschitz on $(\mathcal{G}, d)$ and $Q$ has cells of diameter $\tau$, then $\Delta_{\mathrm{q}} \le L_\ell L_h L_\phi \tau$.

**Proof.** We start from the risk decomposition by code:

$$\varepsilon_t(f) - \varepsilon_s(f) = \sum_{k=1}^{K} \pi_t(k)\left(\mathbb{E}_{\mathbb{P}_t^k}\ell(h(c_k), y) - \mathbb{E}_{\mathbb{P}_s^k}\ell(h(c_k), y)\right) + \sum_{k=1}^{K}(\pi_t(k) - \pi_s(k))\,\mathbb{E}_{\mathbb{P}_s^k}\ell(h(c_k), y).$$

(11)

The second sum is bounded by $\mathrm{TV}(\pi_t, \pi_s)$ since $\ell \in [0, 1]$.

*Step 1 (replace $Q \circ \phi$ by $\phi$ with distortion).* Insert and subtract $\ell(h(\phi(g)), y)$ inside each conditional expectation. By the definition of $\Delta_{\mathrm{q}}$ and the misassignment indicator $M(g)$,

$$\left|\mathbb{E}_{\mathbb{P}_\alpha^k}\ell(h(c_k), y) - \mathbb{E}_{\mathbb{P}_\alpha^k}\ell(h(\phi(g)), y)\right| \le \Delta_{\mathrm{q}} + \mathbb{P}_{\mathbb{P}_\alpha^k}[M(g)] \le \Delta_{\mathrm{q}} + p_{\mathrm{mis}}.$$

Applying to $\alpha \in \{s, t\}$ and summing, we accrue an additive $2(\Delta_{\mathrm{q}} + p_{\mathrm{mis}})$; absorb constants to keep a single $(\Delta_{\mathrm{q}} + p_{\mathrm{mis}})$ term.

*Step 2 (conditional IPM bound).* Define $F_k$ as the function class $\{(g, y) \mapsto \ell(h(\cdot), y)$ restricted to code $k\}$.

Feature view. Assume $u \mapsto \ell(h(u), y)$ is $L_u$-Lipschitz, uniformly in $y$. By Kantorovich–Rubinstein duality,

$$\left|\mathbb{E}_{\mathbb{P}_t^k}\ell(h(\phi(g)), y) - \mathbb{E}_{\mathbb{P}_s^k}\ell(h(\phi(g)), y)\right| \le L_u\, W_1\left(\mathcal{L}_t(S(u) \mid k), \mathcal{L}_s(S(u) \mid k)\right) = L_u\, \Delta_k^{\mathrm{feat}}.$$

Structure view. Let $\mathcal{H}$ be the RKHS with kernel $k(\cdot, \cdot)$ and unit ball $\mathbb{B}_{\mathcal{H}}$. Assume the conditional expectation functional over $\psi(g)$ lies in $C\mathbb{B}_{\mathcal{H}}$: $\ell(h(\psi(g)), y) = \langle f_k, \psi(g)\rangle_{\mathcal{H}}$ with $\|f_k\|_{\mathcal{H}} \le C$. Then by the MMD IPM property,

$$\left|\mathbb{E}_{\mathbb{P}_t^k}\ell(h(\phi(g)), y) - \mathbb{E}_{\mathbb{P}_s^k}\ell(h(\phi(g)), y)\right| \le C\, \mathrm{MMD}_{\mathcal{H}}\left(\mathcal{L}_t(\psi \mid k), \mathcal{L}_s(\psi \mid k)\right) = C\, \Delta_k^{\mathrm{struct}}.$$

Thus, in either view,

$$\left|\mathbb{E}_{\mathbb{P}_t^k}\ell(h(\phi(g)), y) - \mathbb{E}_{\mathbb{P}_s^k}\ell(h(\phi(g)), y)\right| \le \Gamma(\Delta_k).$$

Multiply by $\pi_t(k)$ and sum over $k$ to control the first sum in (11).

*Step 3 (finite-sample estimation of conditional IPMs).* Let $\widehat{\Delta}_k$ be an empirical estimator based on $n_t(k)$ and $n_s(k)$ samples in code $k$.

Feature view. Assume $S(u)$ is sub-Gaussian with parameter $\sigma^2$ and bounded support radius $R$ (w.l.o.g. by truncation). Then standard Wasserstein concentration (e.g., Bobkov–Götze type or transportation inequalities) yields, for each $k$ and any $\eta > 0$, with probability $\ge 1 - \eta$,

$$\left|\widehat{W}_1(\widehat{\mathbb{P}}_t^k, \widehat{\mathbb{P}}_s^k) - W_1(\mathbb{P}_t^k, \mathbb{P}_s^k)\right| \le C_1\sigma\left(\sqrt{\tfrac{1}{n_t(k)}} + \sqrt{\tfrac{1}{n_s(k)}}\right) + C_1'\sqrt{\tfrac{\log(1/\eta)}{\min\{n_t(k), n_s(k)\}}}.$$

A union bound over $k$ with $\eta = \delta/(2K)$ gives the displayed $c_1$ term.

Structure view. For bounded kernels, MMD admits sub-Gaussian concentration; with $k(x, x) \le B^2$,

$$\left|\widehat{\mathrm{MMD}}_{\mathcal{H}} - \mathrm{MMD}_{\mathcal{H}}\right| \le C_2 B\left(\sqrt{\tfrac{1}{n_t(k)}} + \sqrt{\tfrac{1}{n_s(k)}}\right) + C_2'\sqrt{\tfrac{\log(1/\eta)}{\min\{n_t(k), n_s(k)\}}}.$$

Apply a union bound across $k$.

*Step 4 (function class complexity for empirical risk plug-in).* If $\varepsilon_\alpha(f)$ is replaced by empirical risks $\hat{\varepsilon}_\alpha(f)$ in (11) to obtain data-driven guarantees, standard symmetrization yields

$$\mathbb{E}\left[\sup_{f \in \mathcal{F}} |\varepsilon_\alpha(f) - \hat{\varepsilon}_\alpha(f)|\right] \le c\, \mathfrak{R}_{n_\alpha}(\mathcal{F}),$$

and concentration around the mean (e.g., Bousquet inequality) adds a term $O(\sqrt{\log(1/\delta)/n_\alpha})$. Since $\mathcal{F}$ is the composition of Lipschitz $h, \ell$ with $\phi$ and either $u$ or $\psi$, $\mathfrak{R}_n(\mathcal{F})$ inherits Lipschitz contractions.

Collecting all pieces completes the proof. □

## E  RESULTS ON VQ TOKENIZER

In this section, we repeat the experiments in the main text and report the results in Figure 6, 7 and 8. Overall, we get similar observations as on RVQ, further supporting our conclusions.

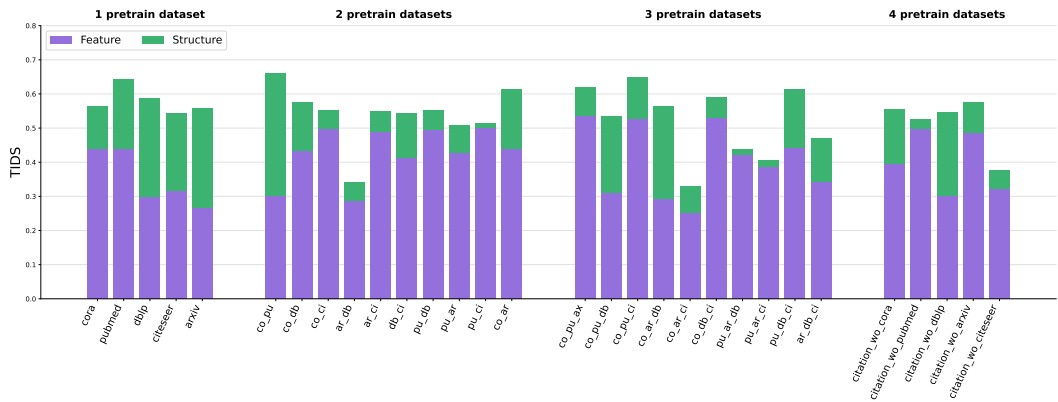

Figure 6: The distributions of GTID of VQ models on citation datasets.

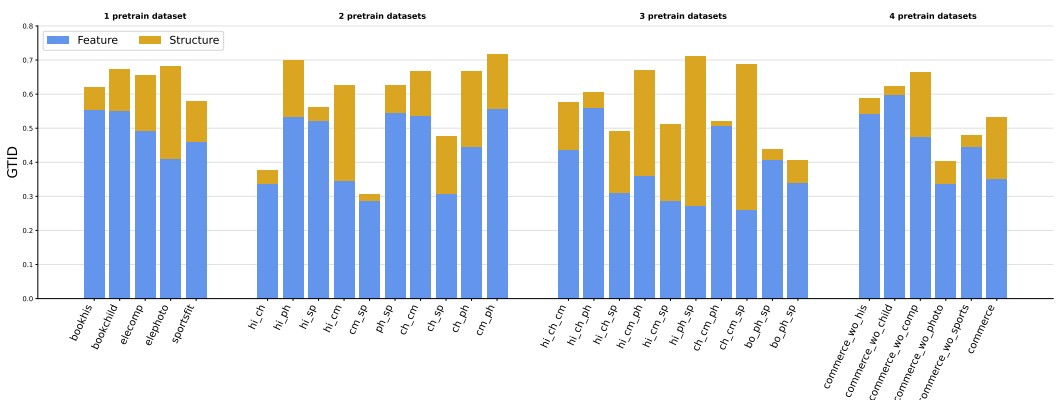

Figure 7: The distributions of GTID of VQ models on e-commerce datasets.

## F  RESULTS ON MORE DOMAINS AND TASKS

We have added experiments on ten additional datasets for two new tasks: link prediction and graph classification. Specifically, we evaluate our method on five knowledge graph datasets for the link prediction task and five molecule datasets for the graph classification task. The details of these datasets are provided in Tables 2 and 3. And the results are shown in Figure 9,10, 11 and 12. Overall, our observations still hold for the new datasets and tasks: the quantization tokenizer cannot effectively capture transferable structural patterns, and the structure GTIDs remain correlated with downstream performance.

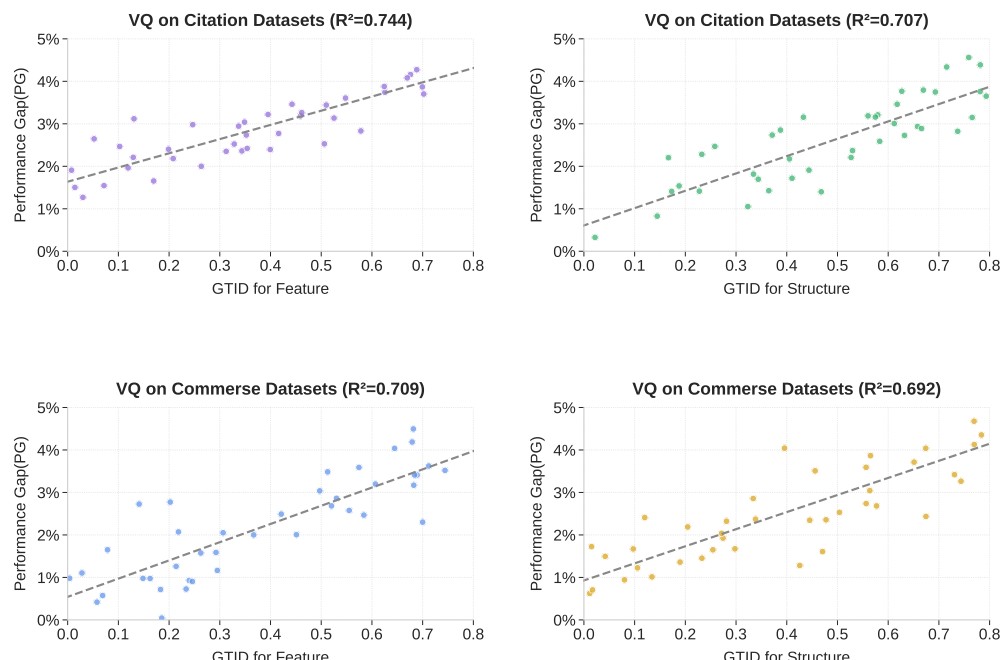

Figure 8: The correlation between GTID and the VQ model performance gap. We observe similar phenomena as RVQ tokenizers.

| Dataset | #Nodes | #Train Triples | #Valid Triples | #Test Triples |
|---|---|---|---|---|
| FB15k237 | 14541 | 272115 | 17535 | 20466 |
| CoDEX Medium | 17050 | 185584 | 10310 | 10311 |
| WN18RR | 40943 | 86835 | 3034 | 3134 |
| NELL995 | 74536 | 149678 | 543 | 2818 |
| ConceptNet100k | 78334 | 100000 | 1200 | 1200 |

Table 2: The statistics of knowledge graph datasets.

## G MORE RESULTS ON OTHER GRAPH TOKENIZERS

To provide a more extensive evaluation of our methodology, we additionally compare two more recent graph tokenization methods: $GPM$ Wang et al. (2025a) and $G^2PM$ Wang et al. (2025b). Specifically, we conduct the same set of experiments on the Citation datasets as in Sections 4 and 5. The results are shown in the Figure 13,14, 15 and 16. From these results, we find that GTID remains strongly correlated with the performance gaps. Moreover, structural GTID is still higher than feature GTID. The key difference is that $G^2PM$ exhibits lower structural GTID than $GPM$, and correspondingly achieves better transfer performance. This can be attributed to $G^2PM$'s more advanced pretraining strategy, which combines both feature and structure reconstruction—consistent with the observations in the main text.

## H MORE RESULTS ON THE ENCODER MODELS

To further demonstrate the generality of our methodology, we additionally evaluate it with different encoder backbones. As suggested, we replace the MPNN encoder with two representative graph transformers: Exphormer and GPS. We run the same experiments on the citation datasets. The results are reported in Figure 17,18, 19 and 20. We observe that our main findings remain consistant across these architectures: the GTID-performance gap correlation still holds, and the limitations of graph quantization tokenizers appear for both MPNN- and transformer-based encoders. This suggests that

| Dataset | #Molecules |
|---------|-----------|
| PCBA | 437,929 |
| HIV | 41,127 |
| ChEMBL | 365,065 |
| MUV | 93,087 |
| ToxCast | 8,576 |

Table 3: The statistics of molecule datasets.

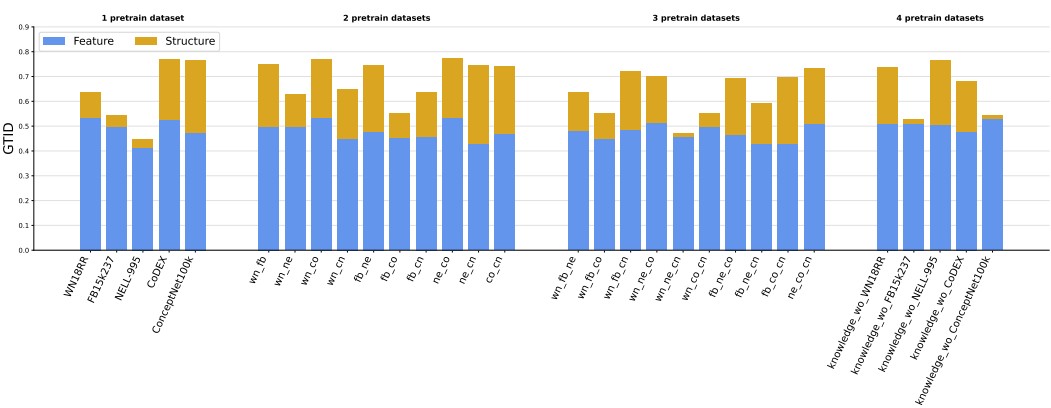

Figure 9: The distributions of GTID of RVQ models on **knowledge graphs** datasets. We observe similar phenomena as in main texts.

the issue is not specific to a single encoder family, but reflects a broader challenge in current graph discretization methods.

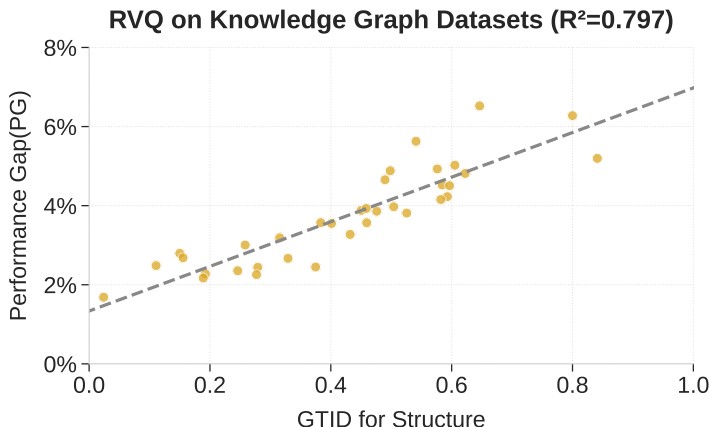

Figure 10: The correlation between GTID and the model performance gap. We observe similar phenomena as in main texts.

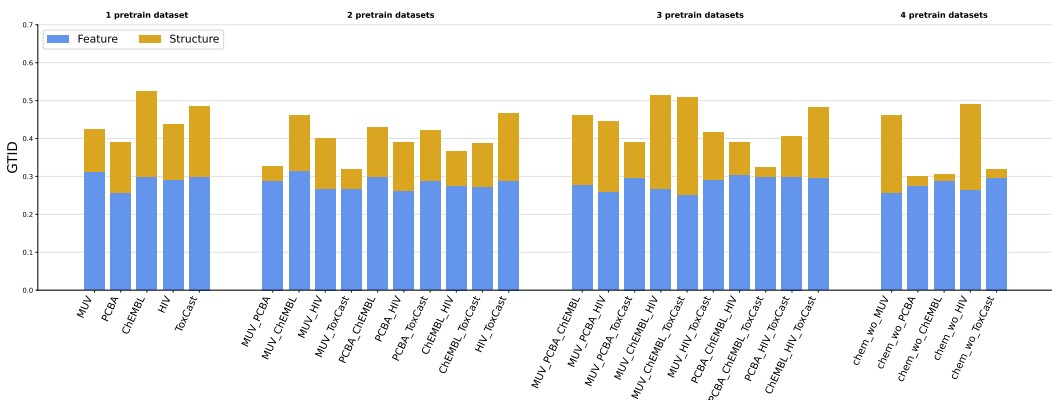

Figure 11: The distributions of GTID of RVQ models on **molecule** datasets. We observe similar phenomena as in main texts.

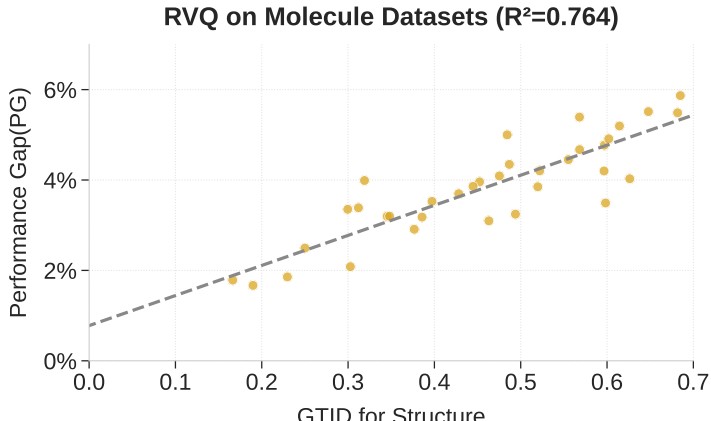

Figure 12: The correlation between GTID and the model performance gap. We observe similar phenomena as in main texts.

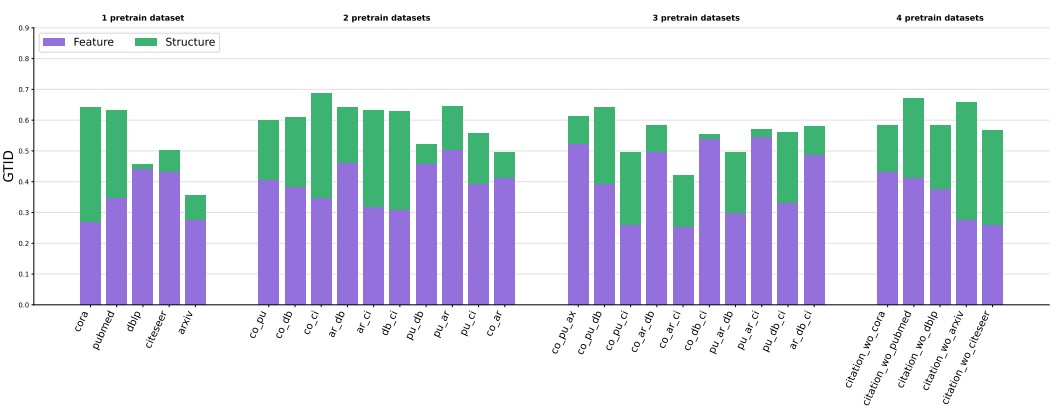

Figure 13: The distributions of GTID of GPM models on citation datasets. We observe similar phenomena as in main texts.

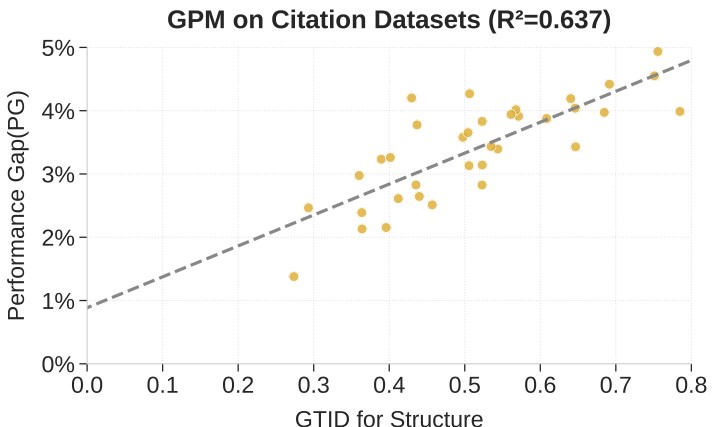

Figure 14: The correlation between GTID and the GPM model performance gap. We observe similar phenomena as in main texts.

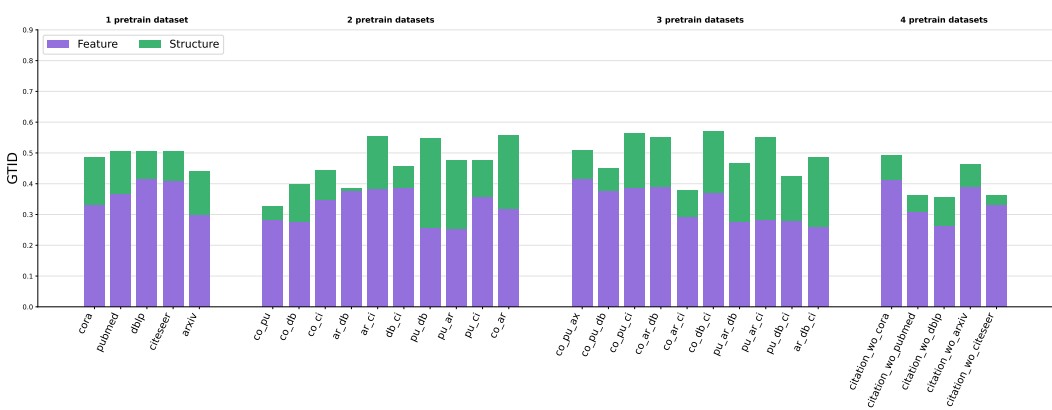

Figure 15: The distributions of GTID of $G^2PM$ models on citation datasets. We observe similar phenomena as in main texts.

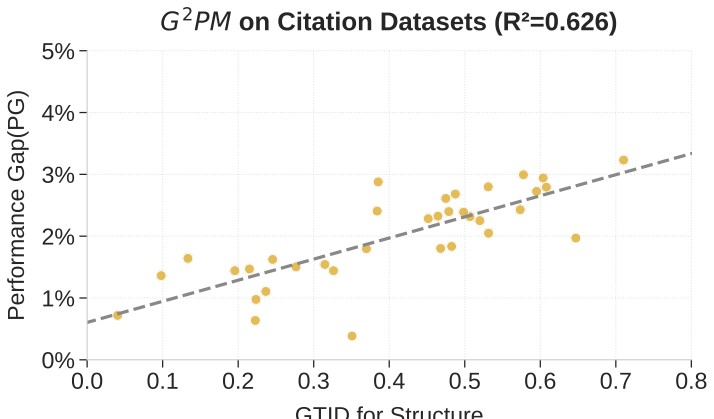

Figure 16: The correlation between GTID and the $G^2PM$ model performance gap. We observe similar phenomena as in main texts.

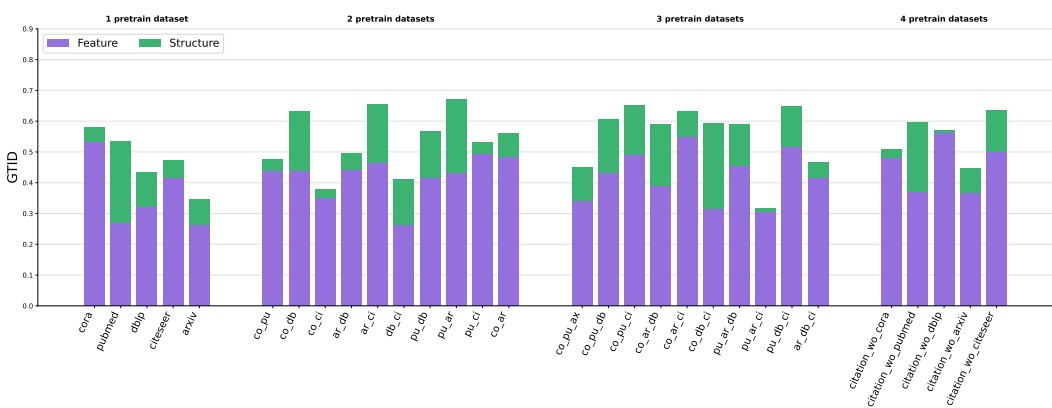

Figure 17: The distributions of GTID of Exphormer+RVQ on citation datasets. We observe similar phenomena as in main texts.

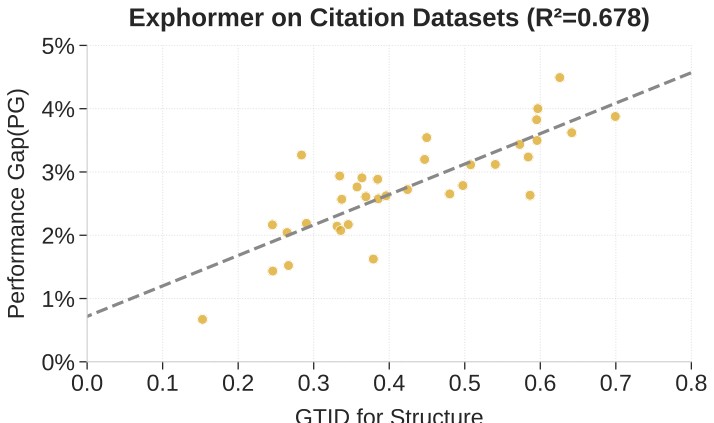

Figure 18: The correlation between GTID and the Exphormer+RVQ model performance gap. We observe similar phenomena as in main texts.

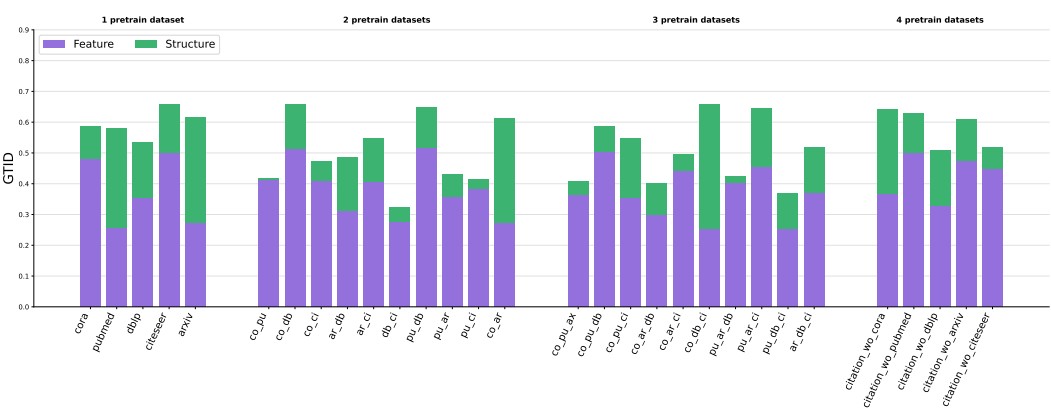

Figure 19: The distributions of GTID of GPS+RVQ models on citation datasets. We observe similar phenomena as in main texts.

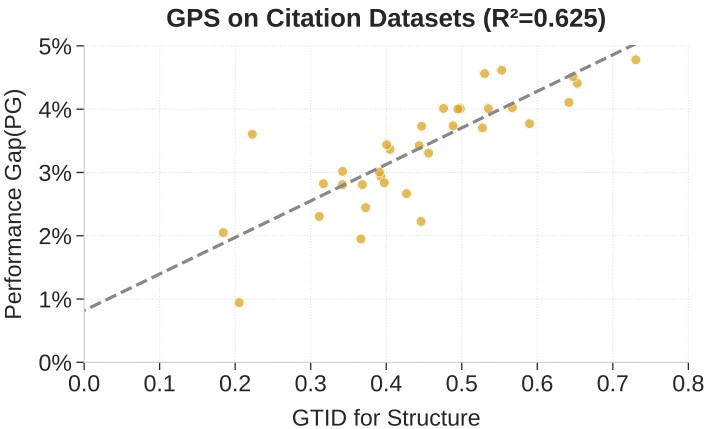

Figure 20: The correlation between GTID and the GPS+RVQ model performance gap. We observe similar phenomena as in main texts.

