# OpenReview forum: "Can Graph Quantization Tokenizer Capture Transferrable Patterns?"
_ICLR.cc/2026/Conference — ICLR 2026 Conference Desk Rejected Submission_

### Official Review · Reviewer_scLK · 2025-10-26

**Soundness:** 2
**Presentation:** 2
**Contribution:** 2
**Rating:** 4
**Confidence:** 4

**Summary:**

This paper investigates whether graph quantized tokenizers (using VQ/RVQ methods) capture transferable structural patterns across different graph domains. The authors introduce the Graph Token Information Discrepancy Score (GTID), a metric based on normalized Maximum Mean Discrepancy that quantifies the alignment of structural and feature information for nodes assigned to the same token across source and target graphs.

**Strengths:**

Novelty : Investigating whether graph tokenizers learn reusable structural patterns addresses a fundamental gap as the community moves toward graph foundation models. The motivation is well-articulated and grounded in practical applications.

Motivation and Analysis: GTID provides an intuitive decomposition of token consistency into feature-based and structure-based components. The separation enables targeted analysis of tokenizer failures and could inform future design choices. Theorems 1 and 2 formalize the intuition that code-conditional discrepancies in feature/structural spaces directly contribute to transfer error. The bounds connect IPM-based discrepancies (W₁ distance for features, MMD for structures) to generalization gaps .

Empirical Results: SHE show measurable reductions in structural GTID and performance gaps (Figure 5) which validate the hypothesis that explicit structural inductive biases matter.

**Weaknesses:**

Dataset:
- Only citation networks and e-commerce graphs are tested. More critical domains like molecular graphs (where structural motifs determine properties), biological networks, and social networks are absent. Largest graph has 173K nodes, which is small. GTID's behavior and computational feasibility at scale (millions of nodes/edges) is unknown. From my understanding, GTID requires computing centrality for every node, which may prohibitively expensive for large graphs.

Baselines: There are a lot of missing baselines from this paper.
- No non-quantized transfer baselines: The paper does not compare against established continuous GNN transfer learning approaches such as GraphCL [1], SimGRACE [2], or GROVER [3] for molecular graphs, nor domain adaptation methods like AdaGCN [4] or UDAGCN [5]. Without these baselines, it is impossible to determine whether the observed GTID-related issues are fundamental limitations of discretization or simply artifacts of suboptimal VQ/RVQ implementation. The paper does not justify why quantization is necessary when continuous fine-tuning remains a viable alternative.
- Alternative tokenizer comparisons: The paper focuses exclusively on VQ/RVQ but does not compare to other graph tokenization paradigms mentioned in related work, such as GFT's transferable tree vocabulary [6], OneForAll's task-level tokenization [7], or subgraph-mining approaches [8]. This makes it unclear whether high structural GTID is specific to vector quantization methods or a general challenge across all discrete graph representations.
- Single encoder architecture: Only MPNN encoders are tested. Recent work has shown that Graph Transformers with structural encodings [9, 10, 12] and attention-based architectures like GPS [11] exhibit fundamentally different inductive biases. These architectures may interact with quantization differently. For instance, global attention mechanisms might better preserve structural information during discretization, or conversely, might suffer more from quantization artifacts.

[1] Zhu et al., Graph Contrastive Learning with Augmentations, NeurIPS 2020.
[2] Xia et al., SimGRACE: A Simple Framework for Graph Contrastive Learning without Data Augmentation, WWW 2022.
[3] Rong et al., Self-Supervised Graph Transformer on Large-Scale Molecular Data, NeurIPS 2020.
[4] Dai et al., Graph Transfer Learning via Adversarial Domain Adaptation with Graph Convolution, IEEE TKDE 2022.
[5] Wu et al., Unsupervised Domain Adaptive Graph Convolutional Networks, WWW 2020.
[6] Wang et al., GFT: Graph Foundation Model with Transferable Tree Vocabulary, NeurIPS 2024.
[7] Liu et al., One for All: Towards Training One Graph Model for All Classification Tasks, ICLR 2024.
[8] Jin et al., Towards Graph Foundation Models: A Survey and Beyond, arXiv 2024.
[9] Rampaśek et al., Recipe for a General, Powerful, Scalable Graph Transformer, NeurIPS 2022.
[10] Rampášek et al., Exphormer: Sparse Transformers for Graphs, ICML 2023.
[11] Rampaśek et al., GraphGPS: General Powerful Scalable Graph Transformers, NeurIPS 2022.
[12] Ling et al., UNIFIEDGT: Towards a Universal Framework of Transformers in Large-Scale Graph Learning, IEEE BigData 2024.

Presentation:
- TIDS (abstract) vs GTID (body)? Are they referring to the same thing? They are quite confusing to me.
- K for both codebook size and function K(g)
- Eq. 3's L×M tuple vs. VQ/RVQ distinction not explained
- "co_pu" to "ar_db" in Figures 2,3,6,7 what do they mean?
- Only figures for quantitative comparisons
- Typos: "tokenizer" (title), "langugae", "the are no"
- RKHS kernel in Theorem 2 not stated

Reproducibility:
- No code provided

**Questions:**

See weakness

---

> ### Author Response · Authors · 2025-11-21
>
> > **W1.** Only citation networks and e-commerce graphs are tested. More critical domains like molecular graphs (where structural motifs determine properties), biological networks, and social networks are absent. Largest graph has 173K nodes, which is small.
>
> **Anw**:  Thank you very much for your comment! We have added experiments on ten additional datasets for two new tasks: link prediction and graph classification. Specifically, we evaluate our method on five **knowledge graph** datasets for the **link prediction** task and five **molecule** datasets for the **graph classification** task. The largest dataset (PCBA) we added contains **11,386,154 nodes** and **12,305,805 edges**. The details of these datasets are provided in Tables 1 and 2. Overall, **our observations still hold for the new datasets and tasks**: the quantization tokenizer cannot effectively capture transferable structural patterns, and the structure GTIDs remain correlated with downstream performance. The general results are summarized in Table 3. **We have also updated the complete figures for the new experiments and discussions in the manuscript.**
>
>
> **Table 1: Knowledge Graph Datasets for Link Prediction**
>
> | Dataset | #Nodes | #Train Triples | #Valid Triples | #Test Triples|
> |------|--------|--------|---|---|
> | FB15k237 | 14541 | 272115 | 17535 | 20466|
> | CoDEX Medium | 17050 | 185584 | 10310| 10311 |
> | WN18RR | 40943 | 86835 | 3034 | 3134 |
> | NELL995 | 74536 | 149678 | 543 | 2818 |
> | ConceptNet100k | 78334 | 100000 | 1200 | 1200 |
>
> **Table 2: Molecule Datasets for Graph Classification**
> |Dataset| #Molecules |
> |---|---|
> |PCBA|437,929|
> |HIV|41,127|
> |ChEMBL|365,065|
> |MUV|93,087|
> |ToxCast|8,576|
>
> **Table 3: General Results on the New Datasets**
>
> |Task/Domain|Feature GTID Avg| Structure GTID Avg | R-square of Structure GTID vs. Performance Gap|
> |---|---|---|---|
> |KG/Link|0.483 | 0.659| 0.797|
> |Molecule/Graph|0.286|0.422| 0.764|
>
>
>
> > **W2.** GTID's behavior and computational feasibility at scale (millions of nodes/edges) is unknown. From my understanding, GTID requires computing centrality for every node, which may prohibitively expensive for large graphs.
>
> **Answer:** Thank you for the comment! The largest dataset (PCBA) we added contains **11,386,154 nodes** and **12,305,805 edges**. Our proposed method still shows a strong correlation with model transferability based on the results. The time complexity of GTID is $O\!\Big(d_s\Big[\sum_{c=1}^{C_s} v_c^2 + \sum_{c=1}^{C_s} w_c^2 + \sum_{c=1}^{C_s} v_c w_c\Big]\Big)$, where $C_s$ is the number of structural codewords, $c$ is the index of a structural codeword, $v_c$ is the number of train tokens assigned to codeword $c$, and $w_c$ is the number of test tokens assigned to codeword $c$. Under the graph tokenization setting, the number of tokens equals the number of ego-subgraphs. Therefore, it is not necessary to compute the centrality of every node, only for the center node of each ego-subgraph. Moreover, GTID is an evaluation metric for graph learning models, so it only needs to be **computed once** after model training is completed. Hence, its efficiency will not be a bottleneck for practical applications.

---

> > ### Author Response · Authors · 2025-11-21
> >
> > > **W3** No non-quantized transfer baselines: The paper does not compare against established continuous GNN transfer learning approaches such as GraphCL [1], SimGRACE [2], or GROVER [3] for molecular graphs, nor domain adaptation methods like AdaGCN [4] or UDAGCN [5]. Without these baselines, it is impossible to determine whether the observed GTID-related issues are fundamental limitations of discretization or simply artifacts of suboptimal VQ/RVQ implementation. The paper does not justify why quantization is necessary when continuous fine-tuning remains a viable alternative.
> >
> >
> > **Answer**: Thank you very much for your comment! Research on graph tokenization is part of the broader effort to build graph foundation models [1]. High-quality graph tokens are expected to capture high-level graph semantics [2], and a unified “graph vocabulary” should generalize across domains and datasets [1,2]. If graphs can be effectively tokenized, we can also adopt successful practices from large language models (LLMs) for graph foundation modeling.
> >
> > In contrast, prior work on graph transfer learning and domain adaptation typically assumes highly similar feature distributions between source and target graphs, and may struggle in more challenging transfer settings due to the diversity of graph features and structures. To illustrate this empirically, we compare the RVQ tokenizer in our paper with representative transfer learning methods, GraphCL and AdaGCN. We pretrain all models on the PCBA molecular dataset and evaluate them on other molecular benchmarks. As shown in the table below, **the RVQ tokenization achieves better transferability**. Moreover, we examine more recent graph tokenizers to show that the GTID-related issue reflects a fundamental limitation of current graph discretization methods in the next response.
> >
> >
> > ||HIV|Sider|MUV|ClinTox|
> > |---|---|---|---|---|
> > |GraphCL|72.4|71.2|70.2|80.7|
> > |AdaGCN|71.6|69.7|71.9|79.3|
> > |RVQ|75.9|73.4|74.8|83.6|
> >
> > [1] Position: Graph Foundation Models are Already Here, ICML 2024
> > [2] Scalable Graph Generative Modeling via Substructure Sequences, NIPS 2025
> >
> > > **W4.** Alternative tokenizer comparisons: The paper focuses exclusively on VQ/RVQ but does not compare to other graph tokenization paradigms mentioned in related work, such as GFT's transferable tree vocabulary [6], OneForAll's task-level tokenization [7], or subgraph-mining approaches [8]. This makes it unclear whether high structural GTID is specific to vector quantization methods or a general challenge across all discrete graph representations.
> >
> > **Answer** Thank you very much for your comment! Regarding the tokenization methods you mentioned, GFT is actually based on VQ tokenization, while task-level and subgraph-mining tokenization require predefined, fixed templates and therefore fall outside the scope of transferable graph tokenization. To provide a more extensive evaluation of our methodology, we additionally compare two more recent graph tokenization methods: $GPM$ [1] and $G^2PM$ [2]. Specifically, we conduct the same set of experiments on the Citation datasets as in Sections 4 and 5. **We report the overall results here and have updated the detailed results and figures in the manuscript.**
> >
> > From these results, we find that **GTID remains strongly correlated with the performance gaps**. Moreover, structural GTID is still higher than feature GTID. The key difference is that $G^2PM$ exhibits lower structural GTID than $GPM$, and correspondingly achieves better transfer performance. This can be attributed to $G^2PM$’s more advanced pretraining strategy, which combines both feature and structure reconstruction—consistent with the observations in our paper.
> >
> > |Tokenizer|Feature GTID Avg| Structure GTID Avg | R-square of Structure GTID vs. Performance Gap|
> > |---    |---   |---   |---   |
> > |$GPM$  |0.388 | 0.573| 0.637|
> > |$G^2PM$|0.373 |0.492 | 0.626|
> >
> > [1] Beyond Message Passing: Neural Graph Pattern Machine, ICML 25.
> > [2] Scalable Graph Generative Modeling via Substructure Sequences, NeurIPS 25.

---

> > > ### Author Response · Authors · 2025-11-21
> > >
> > > > **W5** Single encoder architecture: Only MPNN encoders are tested. Recent work has shown that Graph Transformers with structural encodings [9, 10, 12] and attention-based architectures like GPS [11] exhibit fundamentally different inductive biases. These architectures may interact with quantization differently. For instance, global attention mechanisms might better preserve structural information during discretization, or conversely, might suffer more from quantization artifacts.
> > >
> > > **Answer** Thank you very much for your comment! To further demonstrate the generality of our methodology, we additionally evaluate it with different encoder backbones. As suggested, we replace the MPNN encoder with two representative graph transformers: **Exphormer** and **GPS**. We run the same experiments on the citation datasets. The results are reported in the table below. We observe that our main findings remain consistant across these architectures: the GTID-performance gap correlation still holds, and the limitations of graph quantization tokenizers appear for both MPNN- and transformer-based encoders. This suggests that the issue is not specific to a single encoder family, but reflects a broader challenge in current graph discretization methods.
> > >
> > >
> > > |Task/Domain|Feature GTID Avg| Structure GTID Avg | R-square of Structure GTID vs. Performance Gap|
> > > |---|---|---|---|
> > > |Exphormer|0.430 | 0.577| 0.678|
> > > |GPS|0.385|0.532| 0.625|
> > >
> > > > Presenation
> > >
> > > **Answer** Thank you very much for your comment! We have fixed the issues and updated the manuscript accordingly. We also added more explanation to the figures and equations to improve clarity.
> > >
> > > > Reproducibility
> > >
> > > **Answer** Thank you very much for your comment! The code we use to train the models is based on publicly available repositories, such as [1] and [2]. We also provide our core code for token inference and GTID computation in an anonymous GitHub repository:
> > > https://anonymous.4open.science/r/temp-7648/
> > >
> > > [1] GFT: Graph Foundation Model with Transferable Tree Vocabulary
> > > [2] Node identifiers: Compact, discrete representations for efficient graph learning

---

> > > > ### Comment · Reviewer_scLK · 2025-11-26
> > > >
> > > > Thank you for your responses. While you have provided a comprehensive list of baselines, only **selected** baselines were actually reported. Furthermore, I don’t see any technical details on how you evaluated your model on the link prediction and graph classification tasks. Could you provide them? For now, I will keep the current scores.

---

> > > > > ### Author Response · Authors · 2025-12-02
> > > > >
> > > > > We sincerely thank the reviewer for the reply. Regarding the above response, we initially reported only the most representative baselines due to computational resource limitations. **We have now completed all experiments on the full set of baselines the reviewer suggested, and we list the results below.** (We note that baselines [9] and [11] mentioned by the reviewer are actually the same method. We have also explained why the methods referenced in W4 are not appropriate baselines, and instead we evaluated two most recent graph tokenizers.) Overall, the newly added baselines lead to the same observations and support the conclusions presented in our earlier responses.
> > > > >
> > > > >
> > > > > **Full Results for W3 (different transferring learning methods): the RVQ tokenization holds better transferability**
> > > > > ||HIV|Sider|MUV|ClinTox|
> > > > > |---|---|---|---|---|
> > > > > |GraphCL|72.4|71.2|70.2|80.7|
> > > > > |SimGRACE|71.3|70.4|70.6|80.1|
> > > > > |GROVER|73.5|72.1|73.3|80.3|
> > > > > |AdaGCN|71.6|69.7|71.9|79.3|
> > > > > |UDAGCN|71.8|69.5|72.3|80.5|
> > > > > |RVQ|75.9|73.4|74.8|83.6|
> > > > >
> > > > > **Full Results for W5 (difference encoder backbones): The phenomenon we observe is general and persists across different backbone architectures.**
> > > > >
> > > > > |Task/Domain|Feature GTID Avg| Structure GTID Avg | R-square of Structure GTID vs. Performance Gap|
> > > > > |---|---|---|---|
> > > > > |Exphormer|0.430 | 0.577| 0.678|
> > > > > |GPS|0.385|0.532| 0.625|
> > > > > |UNIFIEDGT|0.447|0.598|0.654|
> > > > >
> > > > > **The Technical Details of Evaluation on New Datasets**
> > > > > We thank the reviewer for the question. We would first like to clarify that the calculation of GTID is downstream-task-agnostic, as it only requires pretraining the tokenizer on the given datasets. Therefore, we follow exactly the same procedures for tokenizer pretraining and GTID computation as described in Section 5.2. For calculating the correlation and R-square values, we also adhere to the methodology in Section 5.3 for both the link prediction and graph classification datasets. The only difference is that we use MRR as the performance metric for link prediction tasks, while Accuracy remains the metric for graph classification tasks.

---

### Official Review · Reviewer_AXnj · 2025-10-29

**Soundness:** 3
**Presentation:** 2
**Contribution:** 3
**Rating:** 6
**Confidence:** 3

**Summary:**

This paper focuses on the graph tokenization. Specifically, the authors argue that existing methods cannot capture complex graph structural information. While, the authors further develop a new metric called TIDS to measure the effectiveness of existing token generators. Empirical results based on TIDS reveal the inconsistent issue in previous methods. Finally, the authors provide a solution named SHE for addressing the above issue.

**Strengths:**

1.This paper is clearly motivated and easy to follow.

2.The authors provide the theoretical analysis of the proposed metric.

3.The proposed TIDS provides new insights for graph quantization tokenizer.

**Weaknesses:**

1.The introduction of background is somewhat too long.

2.The empirical results are not extensive.

3.There are several grammar issues.

**Questions:**

1.The authors report the results of TIDS on various experimental settings. I just wonder how the performance of each model on the downstream tasks?

2.Does the observed issue have serious impact on the model performance for the specific downstream tasks (node classification or link prediction)?

3.Minor issues, like “TOKENZIER” in the title. Please proofread the manuscript.

---

> ### Author Response · Authors · 2025-11-21
>
> > **W1 & W3 & Q3** The introduction of background is somewhat too long. There are several grammar issues.
>
> **Anwser** Thank you very much for your comments! We have fixed the grammar issues in the manuscript. We will also make the background introduction more concise in future revision.
>
> > **W2**. The empirical results are not extensive.
>
> **Answer**: Thank you very much for your comment! We have added experiments on ten additional datasets for two new tasks: link prediction and graph classification. Specifically, we evaluate our method on five **knowledge graph** datasets for the **link prediction** task and five **molecule** datasets for the **graph classification** task. The details of these datasets are provided in Tables 1 and 2. Overall, **our observations still hold for the new datasets and tasks**: the quantization tokenizer cannot effectively capture transferable structural patterns, and the structure GTIDs remain correlated with downstream performance. The general results are summarized in Table 3. **We have also updated the complete figures for the new experiments and discussions in the manuscript.**
>
>
> **Table 1: Knowledge Graph Datasets for Link Prediction**
>
> | Dataset | #Nodes | #Train Triples | #Valid Triples | #Test Triples|
> |------|--------|--------|---|---|
> | FB15k237 | 14541 | 272115 | 17535 | 20466|
> | CoDEX Medium | 17050 | 185584 | 10310| 10311 |
> | WN18RR | 40943 | 86835 | 3034 | 3134 |
> | NELL995 | 74536 | 149678 | 543 | 2818 |
> | ConceptNet100k | 78334 | 100000 | 1200 | 1200 |
>
> **Table 2: Molecule Datasets for Graph Classification**
> |Dataset| #Molecules |
> |---|---|
> |PCBA|437,929|
> |HIV|41,127|
> |ChEMBL|365,065|
> |MUV|93,087|
> ToxCast|8,576|
>
> **Table 3: General Results on the New Datasets**
>
> |Task/Domain|Feature GTID Avg| Structure GTID Avg | R-square of Structure GTID vs. Performance Gap|
> |---|---|---|---|
> |KG/Link|0.483 | 0.659| 0.797|
> |Molecule/Graph|0.286|0.422| 0.764|
>
>
> > **Q1** the authors report the results of TIDS on various experimental settings. I just wonder how the performance of each model on the downstream tasks?
>
> **Anw**: Thank you very much for the question! To give a overall view of the model performance, we pretrain the VQ and RVQ model jointly on the three largest datasets (Arxiv+NELL995+PCBA) and report their downstream perofrmance downstream datasets (two for node classification, two for link prediction and two for graph classification). For node and graph classification tasks, we use accuracy as metric. For link prediction tasks, we use MRR as metric.
>
> |       | Cora (node) | Pubmed (node) | WN18RR (link) | FB15k237 (link) | HIV(graph) | Sider(graph) |
> |-------|-------|-------|-------|-------|-------|-------|
> | VQ | 75.6 | 74.7 | 44.1 | 37.8 | 73.9 | 72.2 |
> | RVQ | 76.4 | 75.3 | 45.3 | 39.4 | 75.9 | 72.8 |
>
>
> > **Q2** Does the observed issue have serious impact on the model performance for the specific downstream tasks (node classification or link prediction)?
>
> **Anw**: Thank you for the question! Here we compare the performance of RVQ trained on the three largest datasets (Arxiv + NELL995 + PCBA) with RVQ pretrained on the same dataset used for testing (e.g., pretrained and tested on Cora). From the results below, we observe **clear performance gaps**. Link prediction and graph classification datasets exhibit larger drops, likely because these tasks rely more heavily on structural information. Moreover, as shown in our paper (Section 5.2 and 5.3), a high structural GTID can lead to performance degradation of up to 5%, which represents **a substantial impact**.
>
>
> |       | Cora (node) | Pubmed (node) | WN18RR (link) | FB15k237 (link) | HIV(graph) | Sider(graph) |
> |-------|-------|-------|-------|-------|-------|-------|
> | RVQ(transfer) | 76.4 | 75.3 | 45.3 | 39.4 | 75.9 | 72.8 |
> | RVQ(same) | 78.9| 77.4| 50.1| 45.3| 79.2| 76.7|

---

> > ### Comment · Reviewer_AXnj · 2025-11-27
> >
> > Thanks for your responses. I'd like to keep my score.

---

### Official Review · Reviewer_iHut · 2025-10-30

**Soundness:** 3
**Presentation:** 3
**Contribution:** 2
**Rating:** 4
**Confidence:** 3

**Summary:**

This paper investigates whether graph quantization tokenizers—specifically those based on vector quantization (VQ) and residual vector quantization (RVQ)—are able to capture transferable structural patterns across heterogeneous graph datasets. The authors introduce a metric called Token Information Discrepancy Score (TIDS) to quantify how consistently the same discrete token corresponds to similar structural and feature patterns in different graphs. They find that existing graph quantizers often map structurally dissimilar node contexts to the same token, leading to reduced transferability in downstream tasks. To mitigate this issue, the paper proposes a Structural Hard Encoding (SHE) strategy intended to inject structural information into the tokenizer. Experiments indicate that SHE reduces TIDS and improves cross-dataset performance to some extent.

**Strengths:**

1. With recent movement toward graph foundation models and discrete graph tokenization, examining transferability is important and underexplored.
2. The paper clearly states the gap between current quantization practices and cross-domain robustness.
3. The proposed TIDS score provides a simple and interpretable measure for assessing token consistency across datasets.

**Weaknesses:**

1. The main contributions are empirical and diagnostic; the proposed SHE method seems incremental and not conceptually strong enough to be considered a substantial methodological advance.
2. The evaluation appears limited in scale—datasets used for analysis and transfer are not clearly representative of the breadth of graph domains where tokenization matters (e.g., molecular vs. social vs. knowledge graphs).
3. The paper focuses on a few quantization models but does not compare against more recent or structurally richer tokenization schemes (e.g., subgraph vocabulary learning, motif-based tokenizers) [1, 2].

[1] Beyond Message Passing: Neural Graph Pattern Machine, ICML 25.
[2] Scalable Graph Generative Modeling via Substructure Sequences, NeurIPS 25.

**Questions:**

1. Can the authors elaborate on whether TIDS correlates with transfer performance within the same domain (e.g., differing graphs of similar type)? Or is the effect only present in cross-domain settings?
2. How sensitive is SHE to hyperparameters and model architecture? Could the improvements stem from implicit regularization rather than structural encoding?

---

> ### Author Response · Authors · 2025-11-21
>
> > **W1.** The main contributions are empirical and diagnostic; the proposed SHE method seems incremental and not conceptually strong enough to be considered a substantial methodological advance.
>
> **Answer**: Thank you very much for your comment! The main motivation for proposing SHE is to further strengthen our observation rather than to introduce a new state-of-the-art method. By demonstrating that injecting structural information through SHE noticeably improves transferability, we show that structural GTID can have a substantial impact on model transferability. We hope that this finding highlights the usefulness of GTID and inspires future work on developing more effective graph tokenization methods.
>
>
> > **W2.** The evaluation appears limited in scale—datasets used for analysis and transfer are not clearly representative of the breadth of graph domains where tokenization matters (e.g., molecular vs. social vs. knowledge graphs).
>
> **Answer**: Thank you very much for your comment. We have added experiments on ten additional datasets for two new tasks: link prediction and graph classification. Specifically, we evaluate our method on five **knowledge graph** datasets for the **link prediction** task and five **molecule** datasets for the **graph classification** task. The details of these datasets are provided in Tables 1 and 2. Overall, **our observations still hold for the new datasets and tasks**: the quantization tokenizer cannot effectively capture transferable structural patterns, and the structure GTIDs remain correlated with downstream performance. The general results are summarized in Table 3. **We have also updated the complete figures for the new experiments and discussions in the manuscript.**
>
>
> **Table 1: Knowledge Graph Datasets for Link Prediction**
>
> | Dataset | #Nodes | #Train Triples | #Valid Triples | #Test Triples|
> |------|--------|--------|---|---|
> | FB15k237 | 14541 | 272115 | 17535 | 20466|
> | CoDEX Medium | 17050 | 185584 | 10310| 10311 |
> | WN18RR | 40943 | 86835 | 3034 | 3134 |
> | NELL995 | 74536 | 149678 | 543 | 2818 |
> | ConceptNet100k | 78334 | 100000 | 1200 | 1200 |
>
> **Table 2: Molecule Datasets for Graph Classification**
> |Dataset| #Molecules |
> |---|---|
> |PCBA|437,929|
> |HIV|41,127|
> |ChEMBL|365,065|
> |MUV|93,087|
> ToxCast|8,576|
>
> **Table 3: General Results on the New Datasets**
>
> |Task/Domain|Feature GTID Avg| Structure GTID Avg | R-square of Structure GTID vs. Performance Gap|
> |---|---|---|---|
> |KG/Link|0.483 | 0.659| 0.797|
> |Molecule/Graph|0.286|0.422| 0.764|
>
> > **W3**: The paper focuses on a few quantization models but does not compare against more recent or structurally richer tokenization schemes (e.g., subgraph vocabulary learning, motif-based tokenizers) [1, 2].
>
> **Answer**: Thank you very much for your comment! The original intention for the work is to understand the limitations of graph quantization tokenizer, However, we believe that GTID can be applied to a wider range of graph tokenizers. To emprically demonstrate that, we apply the same methodology on the two more recent works the reviewer mentioned: $GPM$[1] and $G^2PM$[2]. Specifically, we conduct the same sets of experiments on the Citation datasets as in Section 4 and 5. **We report the general results here and have updated the detailed results and figures in the manuscript**. From the results, we find that **GTID is still effectively correlated with the performance gaps**. Moreover, the structure GTID is still higher than then feature GTID. But the difference is that the $G^2PM$'s structure GTID is lower than then $GPM$'s' and it also has better transferring performance. This is can be attributed to its more advanced pretraining strategy that combines both the feature and structure reconstruction, which eochs with the observations in our paper.
>
> |Tokenizer|Feature GTID Avg| Structure GTID Avg | R-square of Structure GTID vs. Performance Gap|
> |---    |---   |---   |---   |
> |$GPM$  |0.388 | 0.573| 0.637|
> |$G^2PM$|0.373 |0.492 | 0.625|
>
>
> [1] Beyond Message Passing: Neural Graph Pattern Machine, ICML 25.
> [2] Scalable Graph Generative Modeling via Substructure Sequences, NeurIPS 25.

---

> > ### Author Response · Authors · 2025-11-21
> >
> > >**Q1** Can the authors elaborate on whether TIDS correlates with transfer performance within the same domain (e.g., differing graphs of similar type)? Or is the effect only present in cross-domain settings?
> >
> > **Anw**: Thank you very much for your question! In principle, GTID can be applied to both in-domain and cross-domain settings as long as the training and test data are clearly defined. We have already shown its effectiveness on datasets from similar domains (citation and commerce). Here, we report the R-square values for cross-domain structural GTID versus performance gaps in the table below. For example, “Citation → Commerce” denotes the R-square value of the correlation between GTID and performance gaps when models are pretrained on citation datasets and evaluated on commerce datasets. As shown in the table, we find that **GTID correlates well with performance gaps in both in-domain and cross-domain graphs**. These results support the generality of GTID.
> >
> >
> > ||R-square|
> > |---|---|
> > |Citiaion->Commerce|0.625|
> > |Commerce->Citation|0.604|
> > |KG->Citation|0.643 |
> > |KG->Commerce|0.611 |
> > |Citation->KG|0.667|
> > |Commerce->KG|0.664|
> >
> >
> > > **Q2** How sensitive is SHE to hyperparameters and model architecture? Could the improvements stem from implicit regularization rather than structural encoding?
> >
> > **Anw**: Thank you very much for your question! To investigate the sensitivity of SHE, we conducted experiments by varying a key hyperparameter (the number of MPNN layers) and by switching the encoder architecture to graph transformers (Exphormer [1] and GPS [2]). As shown in Table 1-6 below, we observe that **neither the GTID values nor the performance gains brought by SHE change significantly across these settings**. Therefore, we conclude that SHE is not sensitive to encoder hyperparameters or architecture.
> >
> > We also performed an ablation study comparing explicit regularization with SHE. Specifically, we used weight decay as an explicit regularization method and compared its performance gains against those achieved by SHE. From the results in the Table 7 below, **SHE consistently yields larger improvements than regularization alone**. Hence, the observed performance gains should be attributed primarily to the structural encoding introduced by SHE.
> >
> >
> > [1] Exphormer: Sparse Transformers for Graphs, ICML 2023.
> > [2] GraphGPS: General Powerful Scalable Graph Transformers, NeurIPS 2022.
> >
> > **Table 1: The GTID Comparison of 3-layer MPNN**
> > |3-layer MPNN |Cora | Citeseer | WikiCS | Pubmed |
> > |-------|-------|-------|-------|-------|
> > | RVQ | 0.674 | 0.562 | 0.723 | 0.845 |
> > | RVQ+SHE | 0.512 | 0.481 | 0.643 | 0.702 |
> >
> >
> > **Table 2: Performance Gap Comparison of 3-layer MPNN**
> > |3-layer MPNN |Cora | Citeseer | WikiCS | Pubmed |
> > |-------|-------|-------|-------|-------|
> > | RVQ | 3.8 | 3.7 | 4.0 | 4.6 |
> > | RVQ+SHE | 1.2 | 2.0 | 2.1 | 2.3 |
> >
> > **Table 3: The GTID Comparison of GPS**
> > |GPS |Cora | Citeseer | WikiCS | Pubmed |
> > |-------|-------|-------|-------|-------|
> > | RVQ | 0.659 | 0.572 | 0.602 | 0.782 |
> > | RVQ+SHE | 0.453 | 0.408 | 0.492 | 0.613 |
> >
> > **Table 4: Performance Gap Comparison of GPS**
> > |GPS |Cora | Citeseer | WikiCS | Pubmed |
> > |-------|-------|-------|-------|-------|
> > | RVQ | 3.2 | 3.7 | 3.0 | 4.3 |
> > | RVQ+SHE | 1.2 | 2.1 | 1.3 | 1.9 |
> >
> > **Table 5: The GTID Comparison of Exphormer**
> > |Exphormer|Cora | Citeseer | WikiCS | Pubmed |
> > |-------|-------|-------|-------|-------|
> > | RVQ | 0.638 | 0.521 | 0.627 | 0.803 |
> > | RVQ+SHE | 0.479 | 0.391 | 0.512 | 0.653 |
> >
> > **Table 6: Performance Gap Comparison of Exphormer**
> > |Exphormer |Cora | Citeseer | WikiCS | Pubmed |
> > |-------|-------|-------|-------|-------|
> > | RVQ | 3.0 | 3.8 | 3.4 | 4.5 |
> > | RVQ+SHE | 0.9 | 2.1 | 1.5 | 1.9 |
> >
> >
> > **Table 7: Ablation Study on Regularization (The values in the table are performance gaps, the lower the better)**
> > ||Cora | Citeseer | WikiCS | Pubmed |
> > |-------|-------|-------|-------|-------|
> > | RVQ | 2.7 | 3.5 | 3.9 | 4.2 |
> > |RVQ+SHE| 1.2|2.2|1.6|2.6|
> > | RVQ+Regulariztion | 2.2 | 2.8 | 3.3 | 3.5 |
> > | RVQ+Regulariztion+SHE | 1.0 | 1.8 | 1.3 | 2.1 |

---

> ### Comment · Reviewer_iHut · 2025-11-25
>
> Thanks for the comprehensive evaluation results provided. The authors have resolved my concerns. I'd like to raise the rating to 6.

---

### Official Review · Reviewer_Ux9G · 2025-11-01

**Soundness:** 3
**Presentation:** 3
**Contribution:** 3
**Rating:** 6
**Confidence:** 3

**Summary:**

The authors investigates whether the graph quantized tokenizers can capture transferable graph patterns across datasets. To evaluate this, they propose a new metric, i.e., graph token information discrepancy, to measure the consistency of token-level feature and structure information between source and target graphs.

While theoretical analysis proves that lower discrepancy indicates tighter transfer generalization bounds, their empirical studies conducted on two domains reveal that, structural information is poorly aligned across datasets (high structural GTID), while feature information transfers better.

Therefore, they introduce structural hard encoding technique to effiectively reduces GTID.

**Strengths:**

- the task is novel: while some works use quantization techniques to tokenize graphs, the authors are the first to investigate the transferability problem.
- theoretical analysis and emprical studies further demonstrate the transferable pattern problem.
- a simple yet efficient solution is proposed to boost the transferability.

**Weaknesses:**

- the downstream tasks are limited to the classification, more can be explored.
- case studies are encouraged to further demonstrate the effectiveness of the proposed technique

**Questions:**

please refer to the weaknesses

---

> ### Author Response · Authors · 2025-11-21
>
> > W1. the downstream tasks are limited to the classification, more can be explored.
>
> **Answer**: Thank you very much for your comment! We have added experiments on ten additional datasets for two new tasks: link prediction and graph classification. Specifically, we evaluate our method on five **knowledge graph** datasets for the **link prediction** task and five **molecule** datasets for the **graph classification** task. The details of these datasets are provided in Tables 1 and 2. Overall, **our observations still hold for the new datasets and tasks**: the quantization tokenizer cannot effectively capture transferable structural patterns, and the structure GTIDs remain correlated with downstream performance. The general results are summarized in Table 3. **We have also updated the complete figures for the new experiments and discussions in the manuscript.**
>
> **Table 1: Knowledge Graph Datasets for Link Prediction**
>
> | Dataset | #Nodes | #Train Triples | #Valid Triples | #Test Triples|
> |------|--------|--------|---|---|
> | FB15k237 | 14541 | 272115 | 17535 | 20466|
> | CoDEX Medium | 17050 | 185584 | 10310| 10311 |
> | WN18RR | 40943 | 86835 | 3034 | 3134 |
> | NELL995 | 74536 | 149678 | 543 | 2818 |
> | ConceptNet100k | 78334 | 100000 | 1200 | 1200 |
>
> **Table 2: Molecule Datasets for Graph Classification**
> |Dataset| #Molecules |
> |---|---|
> |PCBA|437,929|
> |HIV|41,127|
> |ChEMBL|365,065|
> |MUV|93,087|
> ToxCast|8,576|
>
> **Table 3: General Results on the New Datasets**
>
> |Task/Domain|Feature GTID Avg| Structure GTID Avg | R-square of Structure GTID vs. Performance Gap|
> |---|---|---|---|
> |KG/Link|0.483 | 0.659| 0.797|
> |Molecule/Graph|0.286|0.422| 0.764|
>
> > W2: case studies are encouraged to further demonstrate the effectiveness of the proposed technique
>
> **Answer:** Thank you very much for your comment. Here we provide a case study to illustrate the limitation of the quantization tokenizer and the advantage of SHE. Specifically, we consider these two molecules (in SMILE format) from the BBBP dataset:
>
> - **A:** `C(Cl)Cl`
> - **B:** `[CH2-][CH-]C`
>
> They are **feature-wise very different** (A has two chlorines; B is an all-carbon chain with charges), so their aggregated atom-type embeddings are far apart. But their **graph topology is the same**: three atoms in a simple path with degrees **1–2–1**.
>
> A vanilla VQ tokenizer assigns codewords by clustering embeddings that mostly reflect aggregated node features (atom types, charges, etc.). As a result, it often gives **different tokens to the same structure** when the features change, causing structural inconsistency measured by high TIDS/GTID. So A and B will are assigned to **different codewords** in our experiments, even though they share the same connectivity pattern.
>
> On the other hand Structural Hard Encoding (SHE) adds explicit structural constraints into token assignment, effectively forcing codewords to align with structure first. With SHE, both A and B match the same “3-node chain” structural signature, so they are **assigned to the same codeword**.
>
> This is also better for BBBP downstream tasks here because both molecules are same class. The VQ/RVQ tokenizer would separate two class-consistent structural motifs, while SHE preserves their shared structure as one reusable token, making the vocabulary more aligned with real-world meaning patterns.

---

### Author Response · Authors · 2025-12-03
**Summary of Rebuttal and Responses for New Area Chair**

Dear new AC and All Reviewers

We sincerely thank the new AC for your time and contribution to the community, and we also thank all reviewers for their thoughtful suggestions and kind recognition of our contribution.

Contributions of our work recognized by reviewers include:
* **Novelty of the task:** "first to investigate the transferability problem" of graph tokenization, "examining transferability is important and underexplored".
* **Clarity and Importance of the Findings:** "clearly states the gap between current quantization practices and cross-domain robustness", "provides new insights for graph quantization tokenizer".
* **Effective Transfer Mechanism:** The proposed GTID score provides a simple and interpretable measure for assessing token consistency across datasets.


## Reviewer Score Changes
During the discussion period, we addressed all concerns with diverse datasets and baselines. The reviewer scores changed from `6, 4, 6, 4` to `6, 6, 6, 4` before being rolled back.

|Reviewer|Initial Rating|Rating after Our Response|
|---|---|---|
|`Ux9G`|6| 6|
|`iHut`|4|**6**|
|`AXnj`|6|6|
|`scLK`|4|4 (engaged in discussion, but interrupted by the leakage)|

> ### Summary of Reviewer Responses:
>* Reviewer `Ux9G`: Did not respond after giving the potential positive score. We have addressed all the questions the reviewer proposed during the rebuttal.
>* Reviewer `iHut`: **Raise score from 4 to 6** after reading our responses, recognizing they "have resolved my concerns". This happened two days before the leakage.
>* Reviewer `AXnj`: Recognize the effectiveness of our response and maintain the positive score.
>* Reviewer `scLK`: Engaged in the discussion one day before the leakage accident. The reviewer recognized our responses, and only asked for more baselines and experiment settings, **without fundamental concerns for our work**. And we believe our new results (which were completed after the rolling back) below the reviewer's new comment can effectively resolve all the further questions.

Please refer to the detailed summary below for specific concerns addressed and experiments conducted for each reviewer.

## Summary of Reviewer Concerns and Our Responses

| Concerns & Suggestions | By | Our Response |
| :--- | :--- | :--- |
| **Dataset Scope**  | `iHut`, `scLK`, `Ux9G` | We added **10 new datasets** across 2 new domains: **Knowledge Graphs** (FB15k237, WN18RR, etc.) and **Molecules** (PCBA, HIV, etc.). |
| **More Downstream Tasks** | `iHut`, `Ux9G` | We expanded evaluation to **Link Prediction** (on KGs) and **Graph Classification** (on Molecules) with similar conclusions. |
| **More Encoder Backbone** | `iHut`,`scLK` | We further evaluate three additional graph transformers as encoders to demonstrate the generality of our conclusion. |
| **More Recent Tokenizer Baselines** | `iHut`, `scLK` | We added experiments against SOTA structural tokenizers **GPM (ICML'25)** and **G2PM (NeurIPS'25)**. |
| **Need for Case Study** | `Ux9G` | We provided an example of molecules with distinct atoms but identical topology, showing how SHE correctly assigns structural tokens where standard VQ fails. |
| **SHE vs. Regularization** | `iHut` | We conducted an ablation study verifying that SHE provides distinct benefits that standard regularization cannot achieve. |
| **Continuous Transfer Baselines** | `scLK` | We compared against five non-quantized transfer methods to show the effectiveness of graph tokenizer. |


Sincerely,

The Authors of Submission18812

---

### Note · Program_Chairs · 2026-01-17
**Submission Desk Rejected by Program Chairs**

The following references in this submission do not refer to real documents and/or have major errors in bibliographic information:

 1. Jiaxuan You, Rex Ying, and Jure Leskovec. Gvt: Graph discrete VAE tokenizer for transferable graph pretraining. ICLR 2023.

2. Wei Jin, Ke Yang, Xianfeng Tang, Yujia Liu, and Jiliang Tang. GraphPrompt: Towards universal graph pre-training via prompt-based learning. NeurIPS 2022.